# COMBINING INDUCTION AND TRANSDUCTION FOR ABSTRACT REASONING

**Wen-Ding Li**[*1] **Keya Hu**[*2] **Carter Larsen**[1] **Yuqing Wu**[1] **Simon Alford**[1] **Caleb Woo**[1]
**Spencer M. Dunn**[1] **Hao Tang**[1] **Wei-Long Zheng**[2] **Yewen Pu**[†4] **Kevin Ellis**[†1]
[1]Cornell  [2]Shanghai Jiao Tong University  [4]Autodesk  [*]co-leads  [†]co-advising
correspondence: {wl678,kellis}@cornell.edu, hu_keya@sjtu.edu.cn

*Please refer to* **the updated arXiv version**, *which has improved results with more authors.*

## ABSTRACT

When learning an input-output mapping from very few examples, is it better to first infer a latent function that explains the examples, or is it better to directly predict new test outputs, e.g. using a neural network? We study this question on ARC by training neural models for *induction* (inferring latent functions) and *transduction* (directly predicting the test output for a given test input). We train on synthetically generated variations of Python programs that solve ARC training tasks. We find inductive and transductive models solve different kinds of test problems, despite having the same training problems and sharing the same neural architecture: Inductive program synthesis excels at precise computations, and at composing multiple concepts, while transduction succeeds on fuzzier perceptual concepts. Ensembling them approaches human-level performance on ARC.

## 1 INTRODUCTION

Robust generalization from few examples remains one of the most important ways in which human intelligence surpasses AI. Much recent work views this generalization as a form of abstract reasoning: Given just a few training input-outputs $x_{\text{train}}, y_{\text{train}}$, together with a test input $x_{\text{test}}$, the idea is to predict the corresponding test output $y_{\text{test}}$ using reasoning strategies such as analogical reasoning, chain-of-thought, inductive program synthesis, or transductive prediction (Thoms et al., 2023; Wang et al., 2024; Witt et al., 2023; Lee et al., 2024; Tang et al., 2024a; Hocquette & Cropper, 2024; Butt et al., 2024). The Abstraction and Reasoning Corpus (Chollet (2019), henceforth ARC) is a few-shot learning benchmark that tests the ability to rapidly learn a diverse range of new skills, and apply them to new situations. Each ARC task is presented as input-outputs over colored grids, but can engage concepts such as occlusion, pathfinding, collision, symmetry, gravity, bouncing, counting, etc., making ARC essentially a composite of many reasoning datasets, and one of the more interesting unsolved benchmarks that stresses broad-coverage few-shot learning (Figure 1).

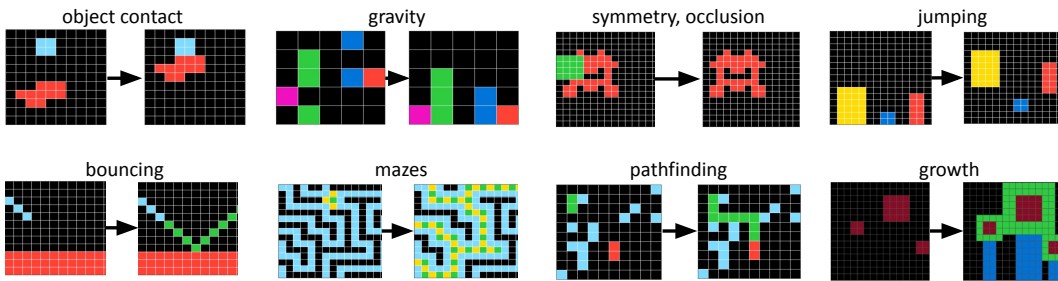

Figure 1: Few-shot learning tasks from the Abstraction and Reasoning Corpus (ARC). Each task typically has 2-5 input-output examples. Here we show just one input-output example per task.

Here we study neural methods for induction and transduction, using few-shot learning problems from ARC as our testbed. *Induction* means first finding a function $f$ where $f(x_{\text{train}}) \approx y_{\text{train}}$, and

then predicting $y_\text{test} = f(x_\text{test})$. *Transduction* instead outputs $y_\text{test}$ without explicit construction of an intermediate function $f$. Intuitively, induction captures the notion that a learner should first explain the training data, then use that explanation to make predictions. Inductive learners can perform better by spending more time optimizing or searching for better explanations, using the training examples $x_\text{train}, y_\text{train}$ to score candidate functions. Transduction instead captures the intuition that the training examples themselves should play a direct role in generating new predictions, and that successful prediction need not require an explicit explanation. See Figure 2.

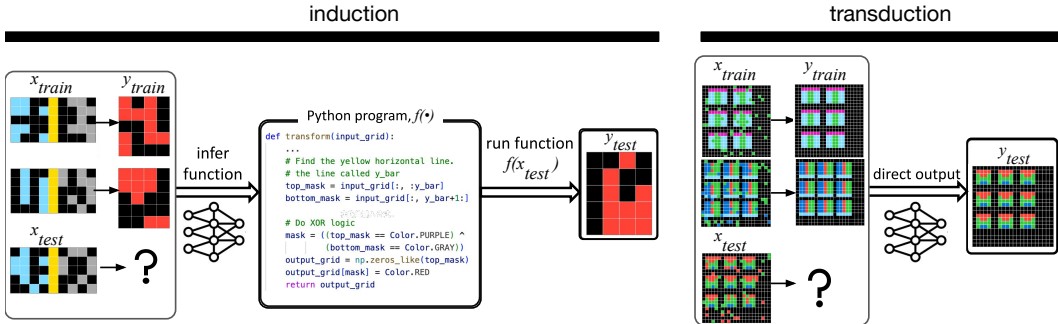

Figure 2: Induction generates an intermediate function $f$ to explain training input-outputs. Transduction directly predicts the test output, for example using a neural network.

We train neural networks for both induction and transduction by generating a large corpus of synthetic problems. We discover that neural models for induction and transduction are strongly complementary. We believe this is surprising: Although any pair of models would generally solve somewhat different problems, usually this can be attributed to different priors, data, or architecture. Instead, we find that, *even controlling for priors, data, and architecture, most problems solved by induction were not solved by transduction, and vice versa.* Moreover, induction and transduction can be trivially ensembled by using induction to generate candidate functions $f$ until either a satisfactory function is found (e.g. $f(x_\text{train}) = y_\text{train}$) or until a test-time compute budget is reached, at which point, transduction kicks in as a fallback: That they are complementary has practical implications.

Our study is tightly linked to program synthesis. We represent functions $f$ as Python code, meaning that induction synthesizes programs. We train transduction models on LLM-produced Python scripts, meaning that transduction is trained on the input-outputs of symbolic code. Although program learning has long been a popular vision of how general AI could work (Solomonoff, 1964; Schmidhuber, 2004; Hutter, 2004), the dominant theory has always been one of explicit code generation (induction), rather than implicitly teaching neural networks to imitate code (transduction). Our work puts this assumption to the test.

Testing these neural methods requires a large dataset of function-learning problems, which is challenging to generate because not only must we make novel functions, but also good inputs to those functions. Consider the transformations in Figure 1: What counts as a good input for one function is unlikely to work for another. To address this challenge, we first generate a deterministic Python function for $f$, and then a probabilistic program for sampling inputs to $f$, finally executing those programs to produce input-outputs. This helps generate inputs that are appropriate for the underlying transformation, and forces $x_\text{train}, y_\text{train}$ to be explainable by a deterministic mapping.

We contribute the following:

1. A study finding that neural models for induction and transduction are strongly complementary, even when trained on the same problems. This contradicts seminal neural program synthesis work (Devlin et al. (2017), which found induction superior), and contradicts the findings of the leading ARC team (Cole et al. (2024), which advocates transduction with test-time training).

2. An automated data generation methodology that starts with 100-160 program solutions for ARC training tasks, and expands them to make 400k new problems paired with Python solutions.

3. A study of how these methods scale. We find performance saturates quickly when increasing manually-labelled data, but scales with compute, both at training and testing time.

4. Analysis of families of problems solved by each approach, and how they compare to humans.

## 2 NEURAL MODELS FOR INDUCTION AND TRANSDUCTION

We consider few-shot supervised learning problems where the learner is trained to map members of an input space $\mathcal{X}$ to output space $\mathcal{Y}$. For $K$-shot learning, we receive $K$ training input-outputs $(\boldsymbol{x}_{\text{train}}, \boldsymbol{y}_{\text{train}}) \in \mathcal{X}^K \times \mathcal{Y}^K$, together with a single test input $x_{\text{test}} \in \mathcal{X}$, and predict $y_{\text{test}} \in \mathcal{Y}$. Our neural models for $K$-shot learning are meta-learned (Mishra et al., 2017, *inter alia.*) using meta-learning data further annotated with a ground-truth function $f : \mathcal{X} \to \mathcal{Y}$, which supervises the induction model. Below we define the training and use of these models.

**Definition: Neural networks for induction and transduction.** A neural network for transduction is a function $\mathsf{t}$ that maps $(\boldsymbol{x}_{\text{train}}, \boldsymbol{y}_{\text{train}}, x_{\text{test}})$ to a distribution over $y_{\text{test}}$, and which has learnable parameters $\theta$. In other words, $\mathsf{t}_\theta : \mathcal{X}^K \times \mathcal{Y}^K \times \mathcal{X} \to \Delta(\mathcal{Y})$, where the notation $\Delta(S)$ means the set of distributions over $S$. We can also write this as a conditional distribution, $\mathsf{t}_\theta(y_{\text{test}}|\boldsymbol{x}_{\text{train}}, \boldsymbol{y}_{\text{train}}, x_{\text{test}})$. A neural network for induction is a function $\mathsf{i}$ that maps $(\boldsymbol{x}_{\text{train}}, \boldsymbol{y}_{\text{train}}, x_{\text{test}})$ to a distribution over functions $f$ that map $\mathcal{X}$ to $\mathcal{Y}$, with learnable parameters $\theta$. In other words, $\mathsf{i}_\theta : \mathcal{X}^K \times \mathcal{Y}^K \times \mathcal{X} \to \Delta(\mathcal{X} \to \mathcal{Y})$, which we can write as a conditional distribution $\mathsf{i}_\theta(f|\boldsymbol{x}_{\text{train}}, \boldsymbol{y}_{\text{train}}, x_{\text{test}})$.

**Training induction and transduction.** Both types of models are trained via meta-learning. We assume a meta-learning dataset $\mathcal{D}$ of few-shot learning problems, each equipped with a ground-truth function $f$ such that $f(x) = y$ for every $x, y$ in $(\boldsymbol{x}_{\text{train}}, \boldsymbol{y}_{\text{train}})$ and $(x_{\text{test}}, y_{\text{test}})$. Inductive and transductive models are meta-trained to minimize the following losses:

$$\text{TRANSDUCTION LOSS} = \mathbb{E}_{(\boldsymbol{x}_{\text{train}}, \boldsymbol{y}_{\text{train}}, x_{\text{test}}, y_{\text{test}}, f) \sim \mathcal{D}} \left[ -\log \mathsf{t}_\theta(y_{\text{test}}|\boldsymbol{x}_{\text{train}}, \boldsymbol{y}_{\text{train}}, x_{\text{test}}) \right] \quad (1)$$

$$\text{INDUCTION LOSS} = \mathbb{E}_{(\boldsymbol{x}_{\text{train}}, \boldsymbol{y}_{\text{train}}, x_{\text{test}}, y_{\text{test}}, f) \sim \mathcal{D}} \left[ -\log \mathsf{i}_\theta(f|\boldsymbol{x}_{\text{train}}, \boldsymbol{y}_{\text{train}}, x_{\text{test}}) \right] \quad (2)$$

**Testing induction and transduction.** After meta-learning the models encounter a test-time few-shot learning task $(\boldsymbol{x}_{\text{train}}, \boldsymbol{y}_{\text{train}}, x_{\text{test}})$. Transductive models predict their most likely output for $y_{\text{test}}$ (approximated via beam search). Inductive models sample a test-time budget of $B$ functions $f_1 \cdots f_B$, which are filtered by $(\boldsymbol{x}_{\text{train}}, \boldsymbol{y}_{\text{train}})$, and finally used to predict $y_{\text{test}} = f(x_{\text{test}})$. Writing $\hat{y}_{\text{test}}$ for the predicted test output:

$$\text{TRANSDUCTION: } \hat{y}_{\text{test}} = \arg\max_{y \in \mathcal{Y}} \mathsf{t}_\theta(y|\boldsymbol{x}_{\text{train}}, \boldsymbol{y}_{\text{train}}, x_{\text{test}}) \quad (3)$$

$$\text{INDUCTION: } \hat{y}_{\text{test}} \sim \text{Uniform}(\mathcal{F}) \quad (4)$$

$$\text{where } \mathcal{F} = \{f_b(x_{\text{test}}) \ : \ \text{for } 1 \le b \le B \text{ if } f_b(\boldsymbol{x}_{\text{train}}) = \boldsymbol{y}_{\text{train}}\}$$

$$f_b \sim \mathsf{i}_\theta(f|\boldsymbol{x}_{\text{train}}, \boldsymbol{y}_{\text{train}}, x_{\text{test}})$$

**Combining induction and transduction.** Induction allows checking candidate hypotheses against the training examples. Therefore, we know when induction has found a plausible solution–but sometimes it fails to find any solution. Transduction has the opposite property: We can't check if its predictions match the training examples, but it always offers a candidate answer. Therefore we ensemble by attempting induction first, then transduction if none of the candidate hypotheses explained the examples:

$$\text{ENSEMBLE: } \hat{y}_{\text{test}} \sim \text{Uniform}(\mathcal{F}) \text{ if } \mathcal{F} \neq \varnothing$$

$$\hat{y}_{\text{test}} = \arg\max_{y \in \mathcal{Y}} \mathsf{t}_\theta(y|\boldsymbol{x}_{\text{train}}, \boldsymbol{y}_{\text{train}}, x_{\text{test}}) \text{ if } \mathcal{F} = \varnothing \quad (5)$$

**Instantiating the framework for ARC.** Every input from $\mathcal{X}$ and output from $\mathcal{Y}$ is a 2D grid ranging from 1–30 pixels per side, with each pixel containing one of ten colors. Because ARC tasks are highly diverse yet typically have an abstract program-like structure, we represent the underlying function $f$ as Python code, which is computationally universal, and so possible in principle of solving any ARC task. Therefore the induction model must generate Python code, so we initialize our models with Llama3.1-8B-instruct (Dubey et al., 2024) because it was pretrained on source code.[1] We encode 2D colored grids as strings using 1 token per pixel, and use newlines to delimit rows (Appendix B.1). We then meta-learn by further fine-tuning Llama3.1-8B-instruct for induction or transduction using a synthetically-generated corpus of problems, described next.

---

[1] Our preliminary experiments suggested Llama3.1-8B-instruct was better than Mistral-7B-v0.3, Qwen2-7B-Instruct, and deepseek-coder-6.7b-instruct

## 3 GENERATING DATASETS FOR INDUCTION AND TRANSDUCTION

Generating ARC-style tasks is challenging because of the diversity of concepts that can occur in ARC. It is also challenging because we need to generate not just a function, and also inputs that serve as good examples for that function.

At a high level, our dataset grows out of 100 manually-written Python programs, each of which both solves a given task (function $f$), and also randomly generates new input grids. We call these 100 manually-written programs **seeds.** Each seed is commented with natural language describing the core intuitions behind the problem. We then use a large language model to mutate and recombine the seeds, producing many thousands of programs (Figure 3).

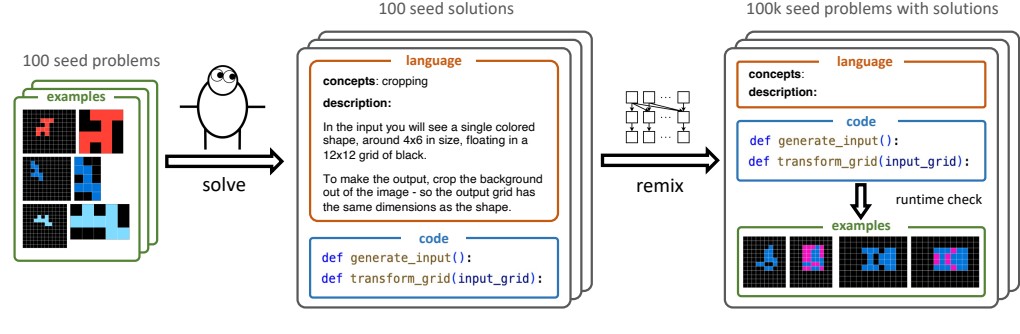

Figure 3: Synthetic data generation pipeline, starting with human-written programs (seeds).

**The structure of seeds.**   Each seed consists of three parts:

1. A **natural language description** of its specific ARC task—including how to solve that task—represented as a Python comment at the top of the seed.
2. A Python function `transform_grid` corresponding to the function $f$ in the manuscript, which maps each input grid of a specific ARC task to its corresponding output grid.
3. A Python function `generate_input`, which takes no arguments, and which randomly generates new inputs to $f$ (new inputs to `transform_grid`).

**Prior knowledge.**   The seeds impart a prior upon the system by demonstrating good programs for solving training tasks. We further codified much of this prior into a Python library containing code that we found useful across many seeds, such as subroutines for generating random sprites, detecting symmetries, or extracting objects (Appendix A.2). Synthetic problems can use that same library.

However, this prior knowledge is different from previous *Domain Specific Languages* for ARC (Butt et al., 2024; Wind, 2020; Ainooson et al., 2023). Domain Specific Languages restrict the class of allowed programs by only allowing stereotyped combinations of domain-specific primitives. We still allow arbitrary Python code, which helps cover the long tail of diverse tasks.

**Remixing the seeds.**   To generate a larger synthetic dataset we "remix" the seeds using LLMs. Each new synthetic ARC problem is generated by a three stage pipeline (Figure 11):

1. A new natural language description is sampled by prompting an LLM with seed natural language descriptions, effectively using in-context learning to recombine and mutate elements of different problems, in the spirit of self-instruct (Wang et al., 2023).
2. Code is generated for that new description via Retrieval Augmented Generation (RAG: Lewis et al. (2020)). Our RAG pipeline retrieves seeds with similar descriptions, and prompts an LLM to generate code for the new description, given the retrieved seeds.
3. The newly created `generate_input` is executed to make inputs, which are passed to `transform_grid` to produce input-output grids.

Figure 4 illustrates example problems generated by our pipeline. Unless otherwise mentioned, we create synthetic datasets with GPT4o-mini and ada-002.

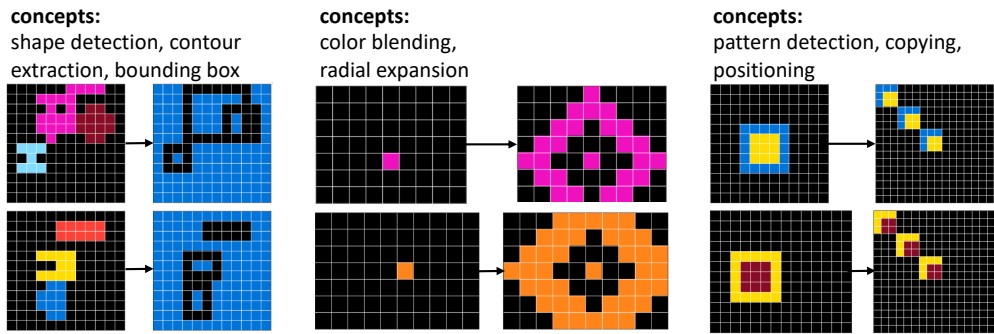

**concepts:**
shape detection, contour extraction, bounding box

**concepts:**
color blending, radial expansion

**concepts:**
pattern detection, copying, positioning

Figure 4: Example synthetic ARC problems generated by our pipeline. Concepts are generated in a comment near the top of the Python script as part of the natural language description of the seed.

## 4 EMPIRICAL STUDY OF INDUCTION AND TRANSDUCTION

We train inductive and transductive models with the goal of understanding (1) how the methods compare; (2) how performance scales with train-time effort; and (3) how performance scales with test-time compute (for induction only, as it allows drawing more samples at test time to improve performance). We report performance on the 400-problem public validation split of ARC, which is harder than the training split. The systems described in this section learn from a 100-problem subset of the training split, specifically problems for which we created seeds.

**Induction and Transduction are strongly complementary.** Despite training on the exact same problems, inductive and transductive models solve different tasks, and neither approach is dramatically more effective than the other. And although these methods have a similar overall solve rate, *most problems solved by induction are not solved by transduction, and vice versa* (Figure 5A).

An alternative explanation is that induction and transduction are not actually complementary, but instead that, having trained two neural networks with different random initializations, they simply solved different problems due to randomness at train or test time. To test this alternative explanation, we trained many models with different random initializations. We find that the problems solved by induction/transduction are surprisingly stable across these different runs (Figure 5B). In other words, some problems are friendlier to induction, and others friendlier to transduction (Figure 6).

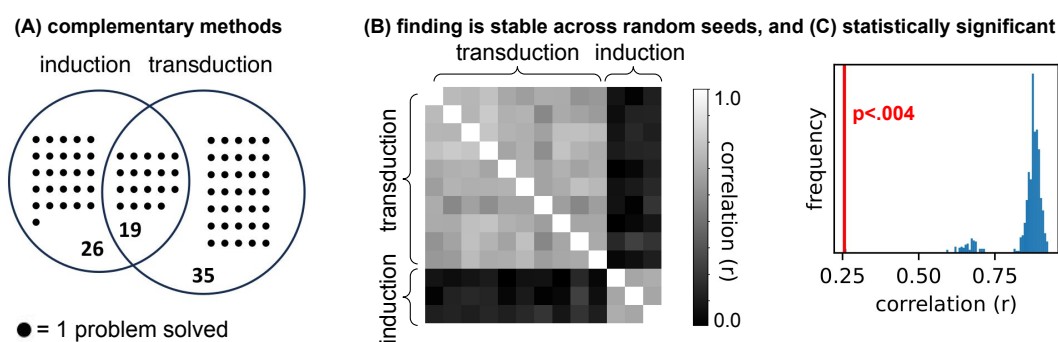

Figure 5: (A) Induction and transduction solve different problems, where solve means predicting the right output given 2 tries. Venn diagram for models trained on 100k synthetic problems generated using gpt4o-mini. (B) Training many models with different random seeds, and then measuring the correlation between solved tasks by different models. Solved tasks strongly correlates with other models of the same class but not the other class. (C) Statistical significance test evaluating the null hypothesis that correlation is independent of whether a model is inductive/transductive.

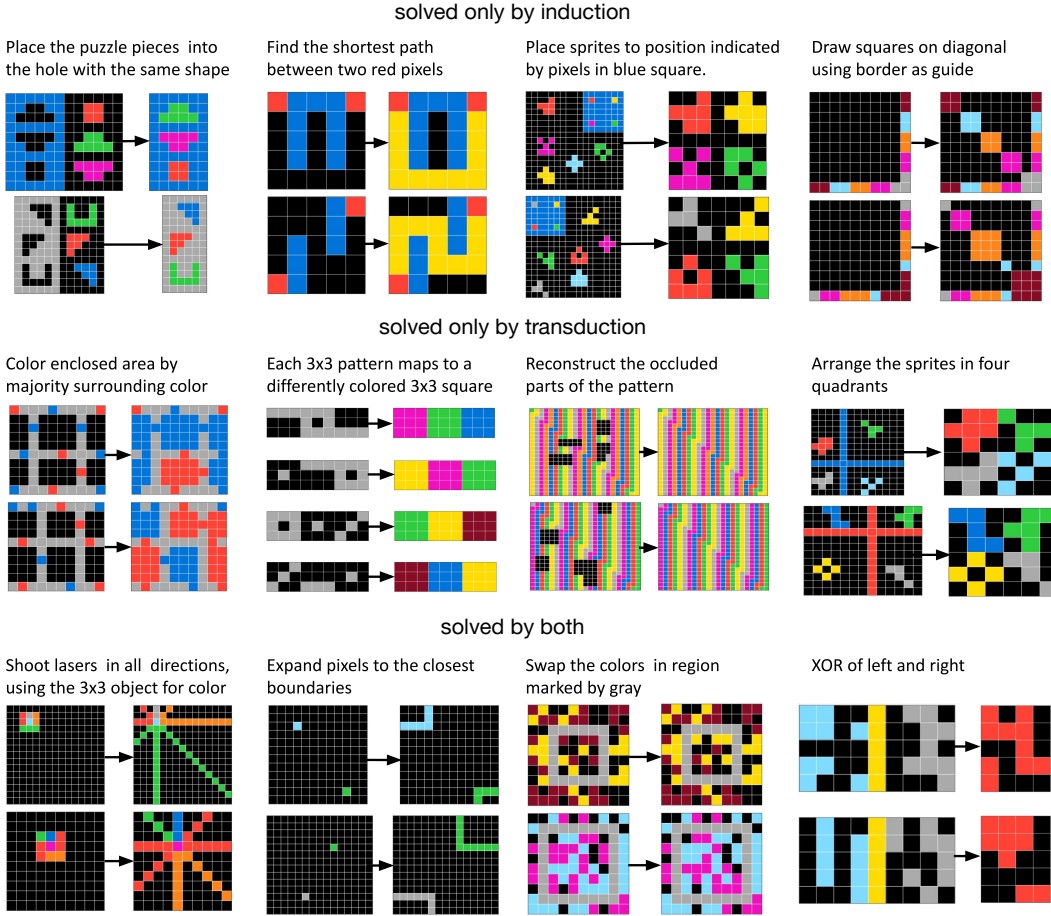

Figure 6: Example tasks solved by induction/transduction/both. See also Appendix C.

**Performance scales with dataset size, but quickly saturates with increasing number of seeds.**
We trained models while systematically varying the number of human-created seeds we use, and
varying the amount of synthetic data generated from those seeds (Figure 7). Performance improves
with increasing training data for fine-tuning (increasing synthetic data), but saturates for increas-
ing quantity of human-created seeds. We conjecture that this saturation occurs because each seed
serves to introduce a few core concepts, and that after enough seeds, essentially all of the important
concepts have been demonstrated. This suggests that, beyond a critical threshold number of seeds,
the method can scale with increasing amounts of compute without demanding further human data
labeling. Looking beyond ARC, this means that our methodology could probably be applied to other
few-shot function-learning problems using a modest amount of manual data labeling.

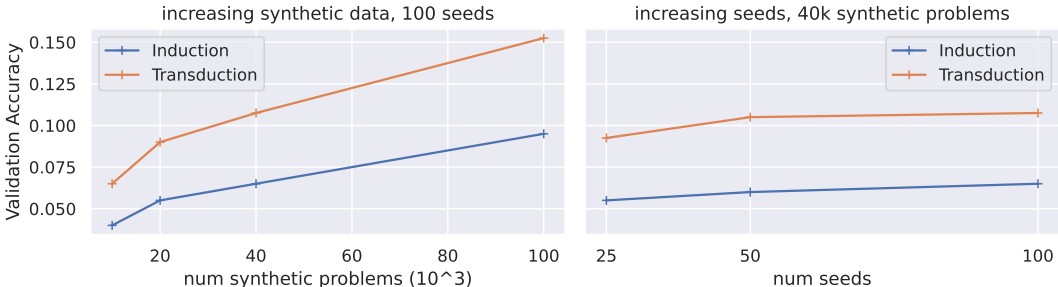

Figure 7: Increased manual human effort (# seeds) does not significantly increase performance, but
increasing compute spent generating synthetic data increases performance of the final model.

**Induction performance scales with test-time compute.** We vary the test-time sampling budget for induction, finding an almost monotonic increase in solve rate (Figure 8, left). In principle, drawing additional samples runs the risk of discovering solutions that are "false-positives," meaning they fit the training examples without correctly predicting the test output. In practice, about 9% of samples that fit the training examples are false-positives. Figure 8 (right) shows that about half of this 9% corresponds to problems where the majority of the probability mass is still placed on the correct output, meaning that a simple majority vote scheme would squash any false positives (e.g. clustering in AlphaCode Li et al. (2022)). Appendix D shows example false positives.

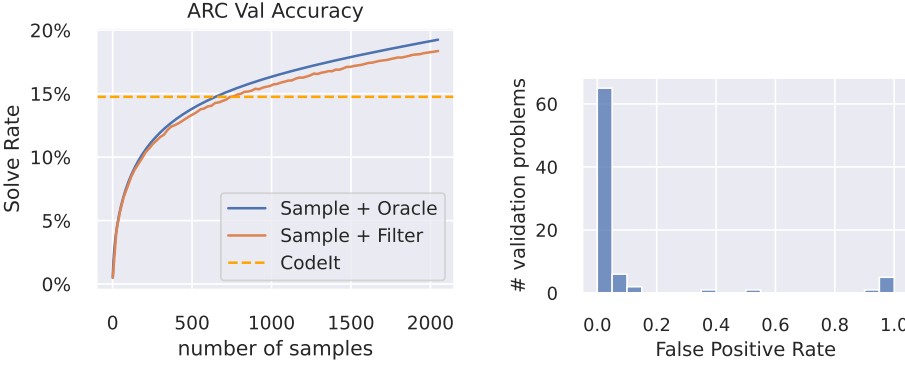

Figure 8: Left: Sample+Oracle assumes an oracle that selects one of the sampled programs. It upper-bounds the accuracy of randomly selecting one program consistent with the training examples (Sample+Filter). Induction model trained with 100k gpt4-description-gpt4omini-codegen data. CodeIt (Butt et al., 2024) is a recent neural program induction model for ARC. Right: Histogram of false positive rate. Of the problems that have false positives, about half have a false positive rate less than 0.5, meaning that most (filtered) samples predict the right test output.

**Stronger LLMs make better synthetic data, and induction is more sensitive to data quality.** To save costs, the previous results all used GPT4o-mini to generate synthetic data. To understand the value of stronger models we generated 100k synthetic problems using GPT4 to generate natural language problem descriptions (but GPT4o-mini still generated the code). The richer and more diverse synthetic problems elicited from GPT4 significantly improved performance, but primarily for induction, while transduction was less sensitive to data quality (Table 1).

Table 1: Val Acc: % validation tasks that are correctly solved in 2 tries.

| System | Val Acc. | Finetuning Data |
|---|---|---|
| Ensemble | 26.50% | |
| Induction, 2048 samples | 18.78% | GPT-4 for generating descriptions, GPT-4o-mini for code |
| Transduction, beam size 20 | 15.25% | |
| Ensemble | 19.50% | |
| Induction, 2048 samples | 11.07% | GPT-4o-mini for generating descriptions and code |
| Transduction, beam size 20 | 13.50% | |

## 5 SCALING OUR METHOD

Motivated by our findings so far, we scaled up our method by producing two larger datasets:

**ARC-Heavy: 200k problems from 160 seeds.** The purpose of ARC-Heavy is to scale our method in an easily reproducible way, while also filling any gaps in its mastery of the training split. We first ran models from Section 4 on the training split to identify 60 problems that they still struggled with, for which we produced 60 new seeds, giving 160 seeds in total. From those seeds we produced 200k synthetic problems, with GPT4 generating natural language descriptions and GPT4o-mini generating the corresponding Python code.

**ARC-Potpourri: 400k problems from heterogeneous sources.** The purpose of ARC-Potpourri is to assemble the biggest dataset that we could, even if it comes from a messy mixture of sources. Starting with ARC-Heavy we added all synthetic data from Section 4. We further added 100k transduction-only training examples from ReARC (Hodel, 2024).

**Test-time improvements.** We improve transduction with test-time training (Sun et al. (2020), abbreviated TTT; Appendix E). We expand our sampling budget to 20k programs.

We call our resulting systems BARC. Table 2 shows the performance of various BARC models. Both transduction and induction are effective, with induction solving slightly more problems, until adding test-time training, after which transduction does slightly better. An ensemble scores 54.375%, surpassing previously published methods. Our best model performs nearly as well as the average human (60.2%) but much worse than the best humans (98%).

Table 2: % validation tasks correctly solved in 2 tries. Human results from LeGris et al. (2024).

| System | Val Acc. | Fair comparison? |
|---|---|---|
| **ARC-Heavy: BARC models** | | |
| Induction, 10k samples, majority vote | 30.50% | — |
| Transduction (no TTT) | 19.25% | — |
| Ensemble (no TTT) | 37.50% | — |
| Transduction (TTT) | 29.75% | — |
| Ensemble (TTT) | 43.25% | — |
| **ARC-Potpourri: BARC models** | | |
| Induction, 20k samples, majority vote | 38.00% | — |
| Transduction (no TTT) | 29.125% | — |
| Ensemble (no TTT) | 47.875% | — |
| Transduction (TTT) | 39.25% | — |
| Ensemble (TTT) | 54.375% | — |
| CodeIt (Butt et al., 2024) | 15% | Yes, only trains on training set |
| Claude-3.5 / Greenblatt (2024) | 21% / 42% | Yes, but closed LLMs at test time |
| Wind (2020) | 39% | No, designed by looking at val set |
| Avg/Best Human | 60.2% / 97.8% | Yes |

# 6    WHICH PROBLEMS ARE EASIER FOR THE MODELS, AND FOR HUMANS?

**Do problems that challenge humans also challenge the model, and vice-versa?** We sort ARC validation problems into 5 equally-sized difficulty classes using data from LeGris et al. (2024). Figure 9 illustrates a peculiar relationship between human and model accuracy: All models surpass human performance on the hardest problems, but underperform on the easiest problems. Because our models train on simple Python programs, this suggests some problems are simple in code and learnable by transformers, but very hard for people—and conversely that people possess priors allowing effortless solution of problems beyond what our Python program generator produces. For problems of typical difficulty, the model roughly tracks human performance, and across all difficulty levels, transduction and induction serve complementary roles, even when augmented with test time training.

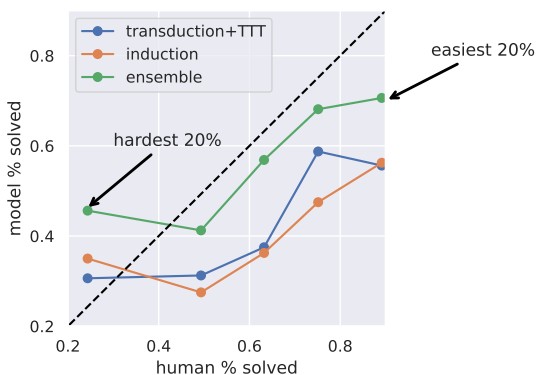

Figure 9: Human vs model performance across 5 difficulty levels. The easiest difficulty level contains problems in the top 20% of human accuracy, and the hardest difficulty level contains the 20% of problems with the lowest human accuracy.

**Which concepts are easier for the models?** We test on ConceptARC (Moskvichev et al., 2023), an alternative ARC-style test-set which classifies its tasks into "concept groups" each exemplifying a single isolated high-level concept such as "sameness" or "above vs below." We use models trained on ARC-Potpourri, finding that specific concept categories are easier for induction or transduction (Figure 10). We find an intuitive division of labor between the two approaches: Concept groups such as counting are best solved with symbolic code, while transduction better handles perceptual processes such as judging whether a shape is more horizontal or more vertical, or more top/bottom.

ConceptARC reveals another dimension along which transduction and induction differ: Because ConceptARC illustrates one concept per problem, there is no need to compose many concepts together. Therefore the induction model, which is uniquely equipped for symbolic composition, loses a key advantage. Transduction has more limited composition capabilities but can instantiate individual concepts in flexible subsymbolic ways, which could explain why it excels on ConceptARC.

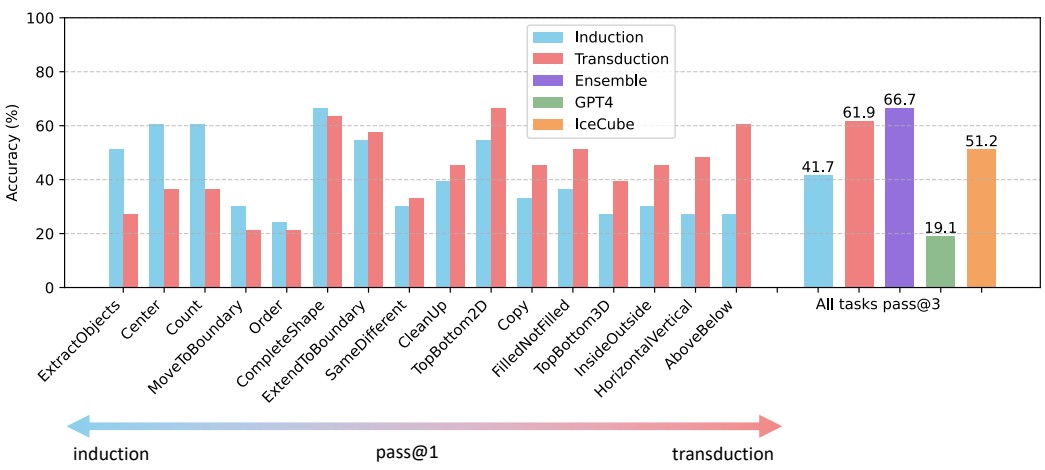

Figure 10: ConceptARC accuracy by concept group. Concept groups sorted left-to-right by ratio of inductive to transductive performance. IceCube is the original ARC Kaggle winner (Wind, 2020). We report pass@3 because Moskvichev et al. (2023) report accuracy given 3 attempts.

## 7 RELATED WORK

**ARC** was originally designed to challenge conventional deep learning and spur progress on alternative paradigms (Chollet, 2019). The first wave of successful approaches used discrete program search over domain-specific programming languages, including the original Kaggle winner (Wind, 2020). These symbolic approaches held their own against GPT-4 (Wang et al., 2024), but have recently been surpassed by transductive architectures using test-time training (Cole et al., 2024), and by LLM-guided program generation (Greenblatt, 2024). ARC has so far resisted conventional neural and symbolic approaches, but is solvable for adult humans, and to some extent, children (LeGris et al., 2024; Opielka et al., 2024).

**Code generation via LLMs** is done in many recent works (Li et al., 2022; Gao et al., 2023; Chen et al., 2021; Austin et al., 2021). We most directly build on Li & Ellis (2024) and Greenblatt (2024). The former fine-tunes LLMs for inductive program synthesis using LLM-generated variations of human-written programs. While there are many technical differences, a key factor is that we generate function inputs by synthesizing an `input_generator` function, rather than have an LLM directly generate possible inputs. This matters because an LLM alone could not generate complex, precisely-correct inputs such as ARC grids. This potentially makes our work applicable to other few-shot generalization problems with complex input-spaces such as webpages, robot planning, etc. Greenblatt (2024) samples many Python programs from GPT4o: Comparable to our induction model, but instead of fine-tuning, it uses prompting. Fine-tuning forced us to create a dataset of new problems, which created the opportunity for exploring transductive models.

Classic work in neural program synthesis has previously compared induction and transduction (RobustFill: Devlin et al. (2017)). We explore here a richer space of functions, reaching a qualitatively different conclusion than RobustFill: Instead of finding transduction inferior to induction, we find them complementary. More broadly, the transductive-inductive divide lies near the heart of supervised learning. Inductive approaches, such as linear regression, first construct a function $f$ where $f(\boldsymbol{x}_{\text{train}}) \approx \boldsymbol{y}_{\text{train}}$, and then predict $y_{\text{test}} = f(x_{\text{test}})$. Transductive approaches, such as Support Vector Machines and In-Context Learning, instead output their predictions by performing direct comparisons with the training data. We use the same neural network architecture and dataset to perform both tasks, allowing a controlled comparison between these paradigms.

## 8 Discussion

**What we learn about robust sample-efficient generalization.** Neither explicit symbolic hypotheses nor implicit neural representations suffice to solve all problems: each has their own domain of applicability, and simply ensembling models specialized in each does not cover all cases. Engineering a more clever neural program search, or training transductive predictors on more data, is unlikely to be fruitful. Instead we need representations irreducible to a purely neural or symbolic form, which intertwine inductive and transductive reasoning. One way of implementing this idea is to do program synthesis within a language whose atomic primitives are non-symbolic, and to pretrain those primitives to encapsulate the basic atoms of core knowledge. While work has taken steps in related directions (Reed & De Freitas, 2015; Alet et al., 2018; Tang & Ellis, 2023; Li et al., 2024), how to engineer and scale this idea remains open.

**To what extent is this methodology applicable beyond ARC?** Few-shot function learning is a very flexible framework, but our particular method is most applicable when the target generalization can be described in symbolic code. As an immediately tangible example, web scraping and other forms of data-munging could fit within our framework. As a more ambitious goal, symbolic code is an especially good medium for defining precise models of how the world works. This is true both within the natural sciences (Schmidt & Lipson, 2009) and also within AI, with examples such as robotic policies (Liang et al., 2023), planners (Wong et al., 2023), and world models more broadly (Das et al., 2023; Tang et al., 2024b; Evans et al., 2021; Liang et al., 2024). These are not the kinds of programs that occur often in LLM pretraining data—so merely prompting is unlikely to perform well—but it is nonetheless feasible to curate around 100 seeds demonstrating what the system should learn.

**Theoretically, induction and transduction should not be so complementary.** Equivalences between induction and transduction are well-know, such as the 'kernel trick' which allows translating parametric function fitting into a transductive problem. Our metalearning models, given infinite metatraining data, should similarly converge because transformers are universal function approximators. That there remains a difference is interesting precisely because it deviates from what one would expect theoretically.

**Limitations.** Our system does not grow more competent at few-shot learning by solving new problems: Instead, it bootstraps from manually encoded knowledge in the seeds, which is transformed into a few-shot learner via an LLM training/inference pipeline. A more compelling approach would be to have the system discover for itself the knowledge that we compiled for it within the seeds, for instance by practicing on training tasks, without supervising on ground truth solutions.

Our work is only evaluated on ARC. However, ARC is designed to contain many concepts and problems embedded within it, so can be viewed as an open-ended composite of different learning problems. Owing to this diversity, it is also notoriously challenging, and has resisted solution despite a series of high-profile competitions. We therefore believe that although evaluating on multiple benchmarks is desirable, ARC is an appropriate benchmark to use as the centerpiece of an experimental evaluation.

**Code & Data Availability.** Our code, data, and model weights are freely available at `https://github.com/xu3kev/BARC`.

**Acknowledgements.** We are grateful for advice from Robert Hawkins regarding the statistical analysis in Figure 5C and for discussions with Weinan Sun about biological learning. This work was partly supported by an NSF CAREER award to K.E.

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

## A  DATA GENERATION TECHNICAL DETAILS

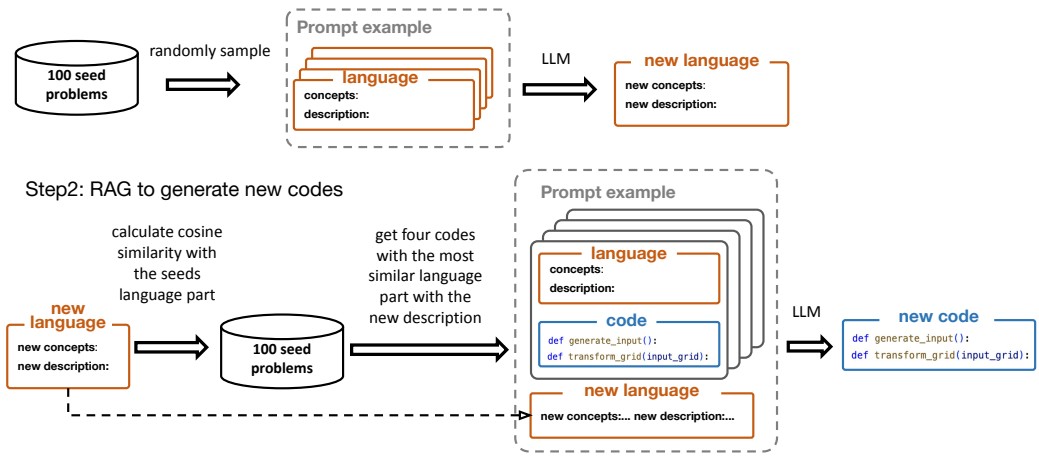

Figure 11: A new natural language description is sampled by prompting an LLM with seed natural language descriptions, effectively using in-context learning to recombine and mutate elements of different problems. Code is generated for that new description via Retrieval Augmented Generation (RAG). Our RAG pipeline retrieves seeds with similar descriptions, and prompts an LLM to generate code for the new description, given the retrieved seeds.

The prompt template for generating natural language descriptions by randomly sampling language descriptions from seed problems is as follows:

```
You've generated these on previous requests:

{examples}

Brainstorm {num_generations} more, using similar thinking:

'''python
# concepts:
# <concepts in your new generation>

# description:
# <description of your new generation>
'''
```

The prompt template for generating Python code from the natural-language descriptions (and similar example code retrieved from the seeds, via RAG) is as follows:

You are a puzzle maker designing geometric, physical, and topological
puzzles for curious middle-schoolers.

Each puzzle consists of uncovering a deterministic rule, pattern, procedure,
 algorithm, or transformation law that maps inputs to outputs.
Both the inputs and outputs are 2D grids of colored pixels. There are 10
colors, but the order of the colors is never relevant to the puzzle.

The middle schoolers are trying to discover this deterministic
transformation, which can be implemented as a Python function called 'main'.
Designing a puzzle involves also creating example inputs, which can be
implemented as a Python function called 'generate_input'. Unlike 'main', the
 'generate_input' function should be stochastic, so that every time you run
it, you get another good example of what the transformation can be applied
to.

Here is a overview of the puzzle you are designing:

{description}

Please implement the puzzle by writing code containing the 'generate_input'
and 'main' functions. Use the following standard library ('common.py'):

```python
{common_lib}
```

Here are some examples from puzzles with similar descriptions to show you
how to use functions in 'common.py':

{examples}

Your task is to implement the puzzle, following these steps:

1. Inspect the example puzzle implementations, making note of the functions
used and the physical/geometric/topological/logical details
2. Inspect the new puzzle's description
3. Brainstorm a possible implementation for the new puzzle
4. Generate a code block formatted like the earlier examples with a comment
starting '# concepts:' listing the concepts and '# description:' describing
the inputs and transformation from the given description.

Be sure to make the transformation 'main' deterministic. Follow the
description closely.

**Execution and Filtering of Generated Problems**    We heuristically filter problems to improve the quality of data based on the following criteria:

- The generator and transformation functions can be executed, producing at least 4 input-output grids examples.

- Transformation being deterministic: We check for consistency by running the functions with different random seeds and filter out those with non-deterministic outputs.

- Appropriate grid sizes: We remove input-output grids with height or width larger than 30, aligning with grid sizes in ARC

- Color permutation check: Since we use numpy arrays with integers 0-9 to represent colors, we want to ensure transformations don't rely on arithmetic operations of these integers. We filter this by checking if input-output remains consistent when permuting the underlying color-number mapping.

- Removal of problems with all trivial identity input-output examples.

A.1 SEED EXAMPLES

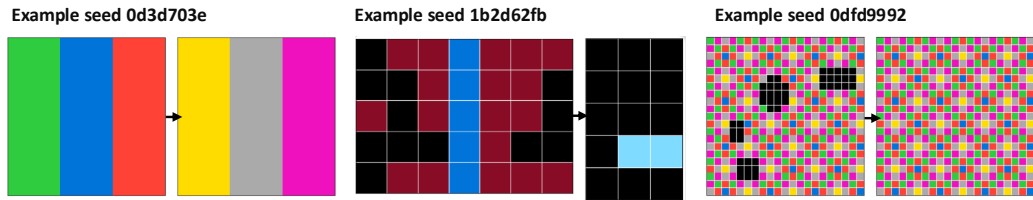

Figure 12: Three seed examples

```python
"""============ problem id: 0d3d703e ============"""
from common import *

import numpy as np
from typing import *

# concepts:
# color mapping

# description:
# The input is a grid where each column is of the same color.
# To make the output, change each color according to the following
                                    mapping:
# green -> yellow, blue -> gray, red -> pink, teal -> maroon, yellow ->
                                    green, gray -> blue, pink -> red,
                                    maroon -> teal

def transform_grid(input_grid):
    # Initialize output grid
    output_grid = input_grid.copy()

    # Performs color mapping
    output_grid = np.vectorize(lambda color: color_map.get(color, color))
                                    (output_grid)

    return output_grid

# Constructing the color map
color_map = {Color.GREEN : Color.YELLOW,
             Color.BLUE : Color.GRAY,
             Color.RED : Color.PINK,
             Color.TEAL : Color.MAROON,
             Color.YELLOW : Color.GREEN,
             Color.GRAY : Color.BLUE,
             Color.PINK : Color.RED,
             Color.MAROON : Color.TEAL
            }

def generate_input():
    grid = np.full((3, 3), Color.BLACK)
    for x in range(grid.shape[0]):
        grid[x, :] = random.choice(list(color_map.keys()))
    return grid

"""============ problem id: 1b2d62fb ============"""

import numpy as np
from typing import *
from common import *
```

```python
# concepts:
# boolean logical operations, bitmasks with separator

# description:
# In the input you will see two maroon bitmasks separated by a blue
                                         vertical bar
# To make the output, color teal the pixels that are not set in either
                                         bitmasks (logical NOR)

def transform_grid(input_grid: np.ndarray) -> np.ndarray:
    # Find the blue vertical bar. Vertical means constant X
    for x_bar in range(input_grid.shape[0]):
        if np.all(input_grid[x_bar, :] == Color.BLUE):
            break

    left_mask = input_grid[:x_bar, :]
    right_mask = input_grid[x_bar+1:, :]

    output_grid = np.zeros_like(left_mask)
    output_grid[(left_mask != Color.MAROON) & (right_mask != Color.MAROON
                                      )] = Color.TEAL

    return output_grid

def generate_input() -> np.ndarray:
    # create a pair of equally sized maroon bitmasks
    width, height = np.random.randint(2, 10), np.random.randint(2, 10)

    grid1 = np.zeros((width, height), dtype=int)
    grid2 = np.zeros((width, height), dtype=int)

    for x in range(width):
        for y in range(height):
            grid1[x, y] = np.random.choice([Color.MAROON, Color.BLACK])
            grid2[x, y] = np.random.choice([Color.MAROON, Color.BLACK])

    # create a blue vertical bar
    bar = np.zeros((1, height), dtype=int)
    bar[0, :] = Color.BLUE

    grid = np.concatenate((grid1, bar, grid2), axis=0)

    return grid

"""============ problem id: 0dfd9992 ============"""

from common import *

import numpy as np
from typing import *

# concepts:
# occlusion, translational symmetry

# description:
# In the input you will see a translationally symmetric pattern randomly
                                 occluded by black pixels.
# To make the output, remove the occluding black pixels to reveal the
                                 translationally symmetric pattern.

def transform_grid(input_grid):
    # Plan:
    # 1. Find the translational symmetries
```

```python
    # 2. Reconstruct the sprite by ignoring the black pixels and
                                    exploiting the symmetry

    w, h = input_grid.shape

    # Identify the translational symmetries
    translations = detect_translational_symmetry(input_grid,
                                    ignore_colors=[Color.BLACK])
    assert len(translations) > 0, "No translational symmetry found"

    # Reconstruct the occluded black pixels by replacing them with colors
                                    found in the orbit of the
                                    symmetries
    output_grid = np.copy(input_grid)
    for x in range(w):
        for y in range(h):
            if output_grid[x, y] == Color.BLACK:
                # Use the translational symmetry to fill in the occluded
                                    pixels
                # to do this we compute the ORBIT of the current pixel
                                    under the
                                    translations
                # and take the most common non-black color in the orbit

                # Compute the orbit into the output
                orbit_pixels = orbit(output_grid, x, y, translations)
                orbit_colors = {input_grid[transformed_x, transformed_y]
                                    for transformed_x, transformed_y in
                                    orbit_pixels}

                # occluded by black, so whatever color it is, black doesn
                                    't count
                orbit_colors = orbit_colors - {Color.BLACK}

                # Copy the color
                assert len(orbit_colors) == 1, "Ambiguity: multiple
                                    colors in the orbit"
                output_grid[x, y] = orbit_colors.pop()

    return output_grid

def generate_input():
    # Make a random large canvas
    grid = np.full((np.random.randint(15, 30), np.random.randint(15, 30))
                                    , Color.BLACK)

    # Make the basic sprite
    w, h = random.randint(3, 8), random.randint(3, 8)
    sprite = random_sprite(w, h, density=1, color_palette=Color.NOT_BLACK
                                    )

    # Place the sprite in the canvas
    for x in range(0, grid.shape[0], w):
        for y in range(0, grid.shape[1], h):
            blit_sprite(grid, sprite, x, y)

    # Create random occluders
    n_occluders = random.randint(1, 5)
    for _ in range(n_occluders):
        x, y = random.randint(0, grid.shape[0]), random.randint(0, grid.
                                    shape[1])
        w, h = random.randint(3, 7), random.randint(3, 7)
        occluder_sprite = np.full((w, h), Color.BLACK)
        blit_sprite(grid, occluder_sprite, x, y)
```

```
    return grid
```

## A.2 COMMON LIBRARY

```python
"""Common library for ARC"""

import numpy as np
import random

class Color:
    """
    Enum for colors

    Color.BLACK, Color.BLUE, Color.RED, Color.GREEN, Color.YELLOW,
    Color.GREY, Color.PINK, Color.ORANGE, Color.TEAL, Color.MAROON

    Use Color.ALL_COLORS for 'set' of all possible colors
    Use Color.NOT_BLACK for 'set' of all colors except black

    Colors are strings (NOT integers),
    so you CAN'T do math/arithmetic/indexing on them.
    (The exception is Color.BLACK, which is 0)
    """

def flood_fill(grid, x, y, color, connectivity=4):
    """
    Fill the connected region that contains the point (x, y) with
    the specified color.

    connectivity: 4 or 8, for 4-way or 8-way connectivity.
    8-way counts diagonals as connected,
    4-way only counts cardinal directions as connected.
    """

def draw_line(grid, x, y, end_x=None, end_y=None, length=None, direction=
                                    None,
              color=None, stop_at_color=[]):
    """
    Draws a line starting at (x, y) extending to (end_x, end_y) or
    of the specified length in the specified direction
    Direction should be a vector with elements -1, 0, or 1.
    If length is None, then the line will continue until it hits
    the edge of the grid.

    stop_at_color: optional list of colors that the line should stop at.
    If the line hits a pixel of one of these colors, it will stop.

    Example:
    # blue diagonal line from (0, 0) to (2, 2)
    draw_line(grid, 0, 0, length=3, color=blue, direction=(1, 1))
    draw_line(grid, 0, 0, end_x=2, end_y=2, color=blue)
    """

def find_connected_components(grid, background=Color.BLACK, connectivity=
                                    4,
                              monochromatic=True):
    """
    Find the connected components in the grid.
    Returns a list of connected
    components, where each connected component is a numpy array.

    connectivity: 4 or 8, for 4-way or 8-way connectivity.
    monochromatic: if True, each connected component is assumed to have
    only one color.
    If False, each connected component can include multiple colors.
    """
```

```python
def random_scatter_points(grid, color, density=0.5,
                          background=Color.BLACK):
    """
    Randomly scatter points of the specified color in the grid with
    specified density.
    """

def scale_pattern(pattern, scale_factor):
    """
    Scales the pattern by the specified factor.
    """

def blit_object(grid, obj, background=Color.BLACK):
    """
    Draws an object onto the grid using its current location.

    Example usage:
    blit_object(output_grid, an_object, background=background_color)
    """

def blit_sprite(grid, sprite, x, y, background=Color.BLACK):
    """
    Draws a sprite onto the grid at the specified location.

    Example usage:
    blit_sprite(output_grid, the_sprite, x=x, y=y,
                background=background_color)
    """

def bounding_box(grid, background=Color.BLACK):
    """
    Find the bounding box of the non-background pixels in the grid.
    Returns a tuple (x, y, width, height) of the bounding box.

    Example usage:
    objects = find_connected_components(input_grid, monochromatic=True,
                         background=Color.BLACK, connectivity=8)
    teal_object=[obj for obj in objects if np.any(obj == Color.TEAL)][0]
    teal_x, teal_y, teal_w, teal_h = bounding_box(teal_object)
    """

def object_position(obj, background=Color.BLACK, anchor="upper left"):
    """
    (x,y) position of the provided object.
    By default, the upper left corner.

    anchor: "upper left", "upper right", "lower left", "lower right",
    "center", "upper center", "lower center", "left center", "right
                                  center"

    Example usage:
    x, y = object_position(obj, background=background_color,
                     anchor="upper left")
    middle_x, middle_y = object_position(obj, background=background_color
                                  ,
                                 anchor="center")
    """

def crop(grid, background=Color.BLACK):
    """
    Crop the grid to the smallest bounding box that contains all
    non-background pixels.

    Example usage:
```

```python
    # Extract a sprite from an object
    sprite = crop(an_object, background=background_color)
    """

def translate(obj, x, y, background=Color.BLACK):
    """
    Translate by the vector (x, y). Fills in the new pixels with the
    background color.

    Example usage:
    red_object = ... # extract some object
    shifted_red_object = translate(red_object, x=1, y=1)
    blit_object(output_grid, shifted_red_object,
            background=background_color)
    """

def collision(_=None, object1=None, object2=None, x1=0, y1=0, x2=0, y2=0,
    background=Color.BLACK):
    """
    Check if object1 and object2 collide when object1 is at (x1, y1) and
    object2 is at (x2, y2).

    Example usage:

    # Check if a sprite can be placed onto a grid at (X,Y)
    collision(object1=output_grid, object2=a_sprite, x2=X, y2=Y)

    # Check if two objects collide
    collision(object1=object1, object2=object2,
            x1=X1, y1=Y1, x2=X2, y2=Y2)
    """

def contact(_=None, object1=None, object2=None, x1=0, y1=0, x2=0, y2=0,
            background=Color.BLACK, connectivity=4,):
    """
    Check if object1 and object2 touch each other (have contact)
    when object1 is at (x1, y1) and object2 is at (x2, y2).
    They are touching each other if they share a border, or if they
    overlap.
    Collision implies contact, but contact does not imply collision.

    connectivity: 4 or 8, for 4-way or 8-way connectivity.
        (8-way counts diagonals as touching,
        4-way only counts cardinal directions as touching)

    Example usage:

    # Check if a sprite touches anything if it were to be placed at (X,Y)
    contact(object1=output_grid, object2=a_sprite, x2=X, y2=Y)

    # Check if two objects touch each other
    contact(object1=object1, object2=object2)
    """

def generate_position_has_interval(max_len, position_num, if_padding=
                                    False):
    """
    Generate the position of the lines with random interval.
    """

def random_free_location_for_sprite(grid, sprite, background=Color.BLACK,
                                    border_size=0, padding=0,
                                    padding_connectivity=8):
    """
    Find a random free location for the sprite in the grid
```

```
    Returns a tuple (x, y) of the top-left corner of the sprite in the
    grid, which can be passed to 'blit_sprite'

    border_size: minimum distance from the edge of the grid
    background: color treated as transparent
    padding: if non-zero, the sprite will be padded with a non-background
    color before checking for collision
    padding_connectivity: 4 or 8, for 4-way or 8-way connectivity when
    padding the sprite

    Example usage:
    x, y = random_free_location_for_sprite(grid, sprite, padding=1,
        padding_connectivity=8, border_size=1, background=Color.BLACK)
    # find the location, using generous padding
    assert not collision(object1=grid, object2=sprite, x2=x, y2=y)
    blit_sprite(grid, sprite, x, y)
    """

def object_interior(grid, background=Color.BLACK):
    """
    Computes the interior of the object (including edges)

    returns a new grid of 'bool' where True indicates that the pixel is
    part of the object's interior.

    Example usage:
    interior = object_interior(obj, background=Color.BLACK)
    for x, y in np.argwhere(interior):
        # x,y is either inside the object or at least on its edge
    """

def object_boundary(grid, background=Color.BLACK):
    """
    Computes the boundary of the object (excluding interior)

    returns a new grid of 'bool' where True indicates that the pixel is
    part of the object's boundary.

    Example usage:
    boundary = object_boundary(obj, background=Color.BLACK)
    assert np.all(obj[boundary] != Color.BLACK)
    """

def object_neighbors(grid, background=Color.BLACK, connectivity=4):
    """
    Computes a mask of the points that neighbor or border the object, but
    are not part of the object.

    returns a new grid of 'bool' where True indicates that the pixel is
    part of the object's border neighbors5.

    Example usage:
    neighbors = object_neighbors(obj, background=Color.BLACK)
    assert np.all(obj[neighbors] == Color.BLACK)
    """

class Symmetry:
    """
    Symmetry transformations, which transformed the 2D grid in ways that
    preserve visual structure.
    Returned by 'detect_rotational_symmetry',
    'detect_translational_symmetry', 'detect_mirror_symmetry'.
    """

    def apply(self, x, y, iters=1):
```

```python
        """
        Apply the symmetry transformation to the point (x, y) `iters`
        times.
        Returns the transformed point (x',y')
        """

def orbit(grid, x, y, symmetries):
    """
    Compute the orbit of the point (x, y) under the symmetry
    transformations `symmetries`.
    The orbit is the set of points that the point (x, y) maps to after
    applying the symmetry transformations different numbers of times.
    Returns a list of points in the orbit.

    Example:
    symmetries = detect_rotational_symmetry(input_grid)
    for x, y in np.argwhere(input_grid != Color.BLACK):
        # Compute orbit on to the target grid, which is typically the
        output
        symmetric_points = orbit(output_grid, x, y, symmetries)
        # ... now we do something with them like copy colors or infer
        missing colors
    """

def detect_translational_symmetry(grid, ignore_colors=[Color.BLACK]):
    """
    Finds translational symmetries in a grid.
    Satisfies: grid[x, y] == grid[x + translate_x, y + translate_y] for
    all x, y, as long as neither pixel is in `ignore_colors`.

    Returns a list of Symmetry objects, each representing a different
    translational symmetry.

    Example:
    symmetries = detect_translational_symmetry(grid, ignore_colors=[
                                    occluder_color])
    for x, y in np.argwhere(grid != occluder_color):
        # Compute orbit on to the target grid
        # When copying to an output, this is usually the output grid
        symmetric_points = orbit(grid, x, y, symmetries)
        for x, y in symmetric_points:
            assert grid[x, y] == grid[x, y] or grid[x, y] ==
                                        occluder_color
    """

def detect_mirror_symmetry(grid, ignore_colors=[Color.BLACK]):
    """
    Returns list of mirror symmetries.
    Satisfies: grid[x, y] == grid[2*mirror_x - x, 2*mirror_y - y]
        for all x, y, as long as neither pixel is in `ignore_colors`

    Example:
    symmetries = detect_mirror_symmetry(grid,ignore_colors=[Color.BLACK])
        # ignore_color: In case parts of the object have been removed and
        # occluded by black
    for x, y in np.argwhere(grid != Color.BLACK):
        for sym in symmetries:
            symmetric_x, symmetric_y = sym.apply(x, y)
            assert grid[symmetric_x, symmetric_y] == grid[x, y]
                or grid[symmetric_x, symmetric_y] == Color.BLACK

    If the grid has both horizontal and vertical mirror symmetries,
        the returned list will contain two elements.
    """
```

```python
def detect_rotational_symmetry(grid, ignore_colors=[Color.BLACK]):
    """
    Finds rotational symmetry in a grid, or returns None if no symmetry
                                    is possible.
    Satisfies: grid[x, y] == grid[y - rotate_center_y + rotate_center_x,
                         -x + rotate_center_y + rotate_center_x]
                         # clockwise
               grid[x, y] == grid[-y + rotate_center_y + rotate_center_x,
                         x - rotate_center_y + rotate_center_x]
                         # counterclockwise
               for all x,y, as long as neither pixel is in ignore_colors

    Example:
    sym = detect_rotational_symmetry(grid, ignore_colors=[Color.BLACK])
        # ignore_color: In case parts of the object have been removed and
        # occluded by black
    for x, y in np.argwhere(grid != Color.BLACK):
        rotated_x, rotated_y = sym.apply(x, y, iters=1) # +1 clockwise,
        -1 counterclockwise
        assert grid[rotated_x, rotated_y] == grid[x, y] or
               grid[rotated_x, rotated_y] == Color.BLACK
    print(sym.center_x, sym.center_y) # In case these are needed, they
                                    are floats
    """

def is_contiguous(bitmask, background=Color.BLACK, connectivity=4):
    """
    Check if an array is contiguous.

    background: Color that counts as transparent (default: Color.BLACK)
    connectivity: 4 or 8, for 4-way (only cardinal directions) or
    8-way connectivity (also diagonals) (default: 4)

    Returns True/False
    """

def random_sprite(n, m, density=0.5, symmetry=None, color_palette=None,
                  connectivity=4, background=Color.BLACK):
    """
    Generate a sprite (an object), represented as a numpy array.

    n, m: dimensions of the sprite. If these are lists, then a random
    value will be chosen from the list.
    symmetry: optional type of symmetry to apply to the sprite. Can be
    'horizontal', 'vertical', 'diagonal', 'radial', 'not_symmetric'. If
    None, a random symmetry type will be chosen.
    color_palette: optional list of colors to use in the sprite. If None,
    a random color palette will be chosen.

    Returns an (n,m) NumPy array representing the sprite.
    """

def detect_objects(grid, _=None, predicate=None, background=Color.BLACK,
                   monochromatic=False, connectivity=None,
                   allowed_dimensions=None,
                   colors=None, can_overlap=False):
    """
    Detects and extracts objects from the grid that satisfy custom
    specification.

    predicate:
        a function that takes a candidate object as input and
        returns True if it counts as an object
    background:
```

```
        color treated as transparent
    monochromatic:
        if True, each object is assumed to have only one color
        If False, each object can include multiple colors.
    connectivity:
        4 or 8, for 4-way or 8-way connectivity.
        If None, the connectivity is determined automatically.
    allowed_dimensions:
        a list of tuples (n, m) specifying the allowed dimensions of the
        objects.
        If None, objects of any size are allowed.
    colors:
        a list of colors that the objects are allowed to have.
        If None, objects of any color are allowed.
    can_overlap: if True, objects can overlap.
        If False, objects cannot overlap.

    Returns a list of objects, where each object is a numpy array.
    """
```

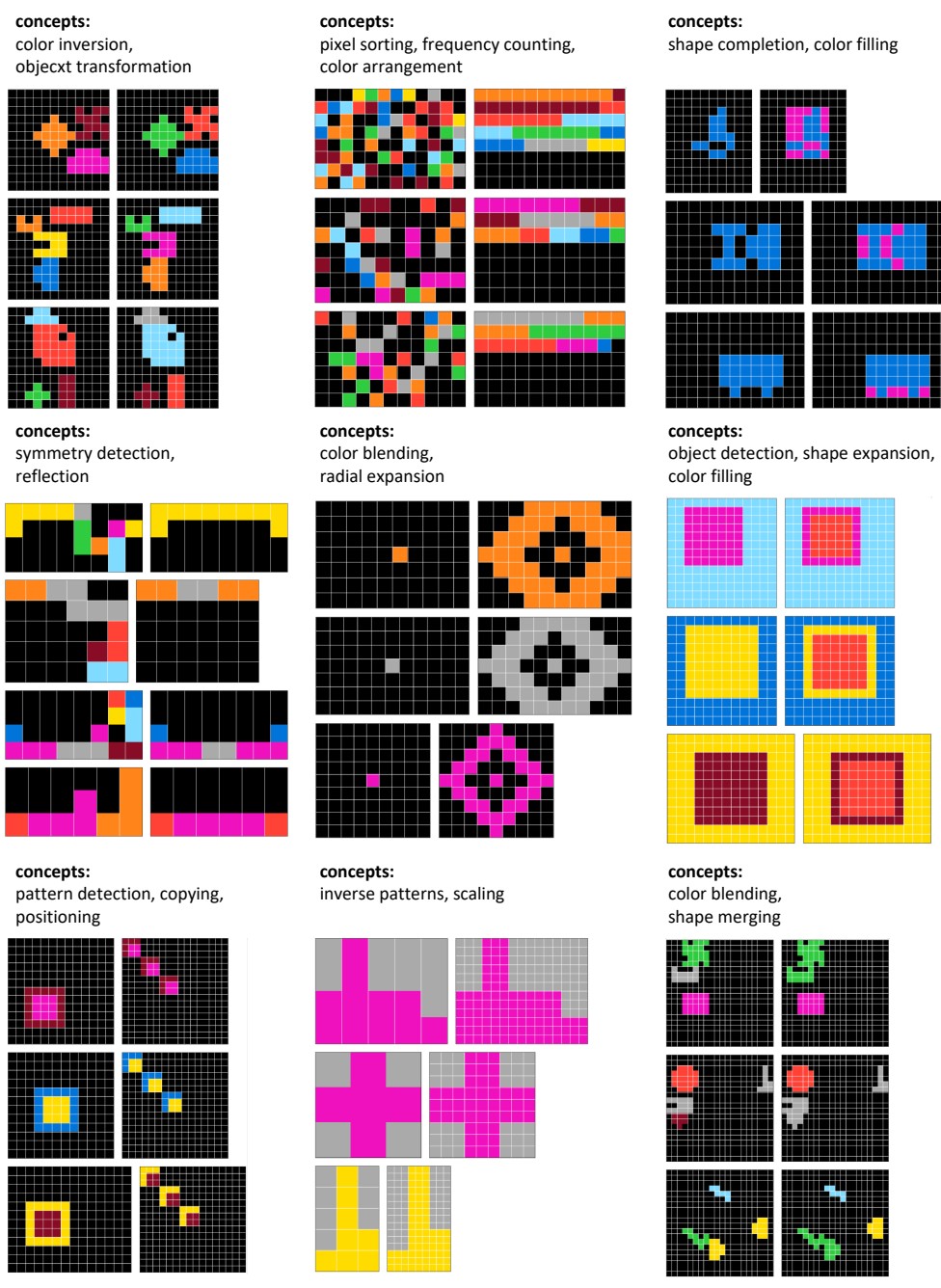

Figure 13: Nine example problems generated automatically by our pipeline.

### A.3 GENERATED ARC EXAMPLES

## B FINE TUNING TRAINING DETAILS

### B.1 PROMPTING THE MODELS

We must include in our prompts for our fine-tuned models the input/output 2D colored grids of each problem. To do this we represent the problem textually by naming the colors one-by-one. We renamed certain colors which were more than one token (e.g., maroon→brown saves 1 token/pixel), and presented the grid as a whitespace-delimited 2D array with newlines delimiting rows. Please see below.

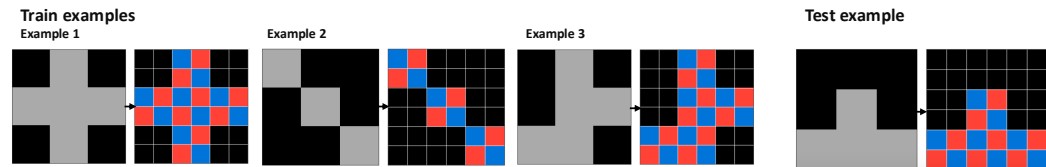

Figure 14: Prompt example illustration

Transduction example:

```
-----  Role: system  --------------------
You are a world-class puzzle solver with exceptional pattern recognition
skills. Your task is to analyze puzzles, spot patterns, and provide direct
solutions.
-----  Role: user  -------------------
Given input-output grid pairs as reference examples, carefully observe the
patterns to predict the output grid for new test input. Each pair follows
the same transformation rule. Grids are 2D arrays represented as strings,
with cells (colors) separated by spaces and rows by newlines.
Here are the input and output grids for the reference examples:
Example 1
Input:
Black Gray Black
Gray Gray Gray
Black Gray Black

Output:
Black Black Blue Red Black Black
Black Black Red Blue Black Black
Blue Red Blue Red Blue Red
Red Blue Red Blue Red Blue
Black Black Blue Red Black Black
Black Black Red Blue Black Black

Example 2
Input:
Gray Black Black
Black Gray Black
Black Black Gray

Output:
Blue Red Black Black Black Black
Red Blue Black Black Black Black
Black Black Blue Red Black Black
Black Black Red Blue Black Black
```

```
Black Black Black Black Blue Red
Black Black Black Black Red Blue

Example 3
Input:
Black Gray Black
Black Gray Gray
Gray Gray Black

Output:
Black Black Blue Red Black Black
Black Black Red Blue Black Black
Black Black Blue Red Blue Red
Black Black Red Blue Red Blue
Blue Red Blue Red Black Black
Red Blue Red Blue Black Black

Here is the input grid for the test example:
Input:
Black Black Black
Black Gray Black
Gray Gray Gray

Directly provide the output grids corresponding to the given test input
grids, based on the patterns observed in the reference examples.
-----  Role: assistant  --------------------
The output grid for the test input grid is:
'''
Black Black Black Black Black Black
Black Black Black Black Black Black
Black Black Blue Red Black Black
Black Black Red Blue Black Black
Blue Red Blue Red Blue Red
Red Blue Red Blue Red Blue
'''
```

Induction example:

```
-----  Role: system  --------------------
You are a world-class puzzle solver with exceptional pattern recognition
skills and expertise in Python programming. Your task is to analyze puzzles
and provide Python solutions.
-----  Role: user  --------------------
Given input-output grid pairs as reference examples, carefully observe the
patterns to predict the output grid for new test input. Each pair follows
the same transformation rule. Grids are 2D arrays represented as strings,
with cells (colors) separated by spaces and rows by newlines.
Here are the input and output grids for the reference examples:
Example 1
Input:
Black Gray Black
Gray Gray Gray
Black Gray Black

Output:
Black Black Blue Red Black Black
Black Black Red Blue Black Black
Blue Red Blue Red Blue Red
Red Blue Red Blue Red Blue
```

```
Black Black Blue Red Black Black
Black Black Red Blue Black Black

Example 2
Input:
Gray Black Black
Black Gray Black
Black Black Gray

Output:
Blue Red Black Black Black Black
Red Blue Black Black Black Black
Black Black Blue Red Black Black
Black Black Red Blue Black Black
Black Black Black Black Blue Red
Black Black Black Black Red Blue

Example 3
Input:
Black Gray Black
Black Gray Gray
Gray Gray Black

Output:
Black Black Blue Red Black Black
Black Black Red Blue Black Black
Black Black Blue Red Blue Red
Black Black Red Blue Red Blue
Blue Red Blue Red Black Black
Red Blue Red Blue Black Black

Here is the input grid for the test example:
Input:
Black Black Black
Black Gray Black
Gray Gray Gray

Write a Python function 'transform' that can convert any given input grid to
 its corresponding output grid based on the pattern observed in the
reference examples.
-----  Role: assistant  --------------------
Let's solve this puzzle using Python code with the common library functions.
 We'll first reason about the problem and then write the code to solve it.
The 'transform' function will take the input grid and return the output grid
. Here is the Python code with the comments describing how to solve the
problem:
'''python
from common import *

import numpy as np
from typing import *

# concepts:
# repetition, diagonal lines, color change

# description:
# In the input you will see a 3x3 grid with a specific diagonal pattern of
gray pixels.
# The output should be the result of repeating this diagonal pattern
horizontally and vertically
```

```
    # to fill a 6x6 grid, changing the color from gray to blue and red in an
    alternating manner.

    def transform(input_grid: np.ndarray) -> np.ndarray:
        # Create a new 6x6 output grid filled with the background color (black)
        output_grid = np.full((6, 6), Color.BLACK)

        # Check for the diagonal pattern in the input grid
        for i in range(3):
            for j in range(3):
                if input_grid[i, j] == Color.GRAY:
                    # Fill the same position in the output grid with alternating
      colors
                    output_grid[i * 2, j * 2] = Color.BLUE
                    output_grid[i * 2, j * 2 + 1] = Color.RED
                    output_grid[i * 2 + 1, j * 2] = Color.RED
                    output_grid[i * 2 + 1, j * 2 + 1] = Color.BLUE

        return output_grid
    ```
```

## C  EXAMPLES OF INDUCTION SOLUTIONS

### C.1  ARC PROBLEM F3CDC58F

```python
from common import *

import numpy as np
from typing import *

# concepts:
# color counting, grid transformation

# description:
# In the input, you will see a grid with several colored squares.
# To make the output, count the number of squares of each color.
# Fill the output grid with bars of corresponding colors.
# The height of each bar corresponds to the number of squares of that
                                    color.

def transform(input_grid: np.ndarray) -> np.ndarray:
    # Create a blank output grid
    output_grid = np.zeros_like(input_grid)

    # Count the number of squares of each color
    color_counts = {color: 0 for color in Color.NOT_BLACK}

    for row in input_grid:
        for cell in row:
            if cell in color_counts:
                color_counts[cell] += 1

    # Define the height of the bars based on the counts
    max_height = 10  # Maximum height of the bars
    for color, count in color_counts.items():
        if count > 0:
            for h in range(min(count, max_height)):
                output_grid[-(h + 1), (color - 1) % len(Color.NOT_BLACK)]
                                        = color  # Fill the
                                        output grid with
```

```
                                                         colors in a bar
                                                         representation

    return output_grid
```

## C.2  ARC PROBLEM BE03B35F

```python
from common import *

import numpy as np
from typing import *

# concepts:
# color detection, pattern extraction, rotation transformation

# description:
# In the input, you will see a 5x5 grid consisting of a colored pattern
#                                 in the top left corner and a 2x2
#                                 square of red pixels in the bottom
#                                 right corner.
# Depending on the color of the top left corner (blue or green), rotate
#                                 the pattern by 90 degrees clockwise
#                                 or counterclockwise.
# The output grid should be the transformed pattern placed in the same
#                                 position as the original top left
#                                 pattern.

def transform(input_grid):
    # Extract the top left 5x5 pattern
    pattern = input_grid[0:5, 0:5]

    # Identify the color in the top left corner
    corner_color = pattern[0, 0]

    # Define rotation based on the corner color
    if corner_color == Color.BLUE:
        transformed_pattern = np.rot90(pattern, k=-1)  # Rotate 90
                                            degrees counterclockwise
    elif corner_color == Color.GREEN:
        transformed_pattern = np.rot90(pattern, k=1)    # Rotate 90
                                            degrees clockwise
    else:
        raise ValueError("The corner color must be either BLUE or GREEN."
                                            )

    # Create the output grid
    output_grid = np.full((2, 2), Color.BLACK)  # Initialize a small grid
    output_grid = blit_sprite(output_grid, transformed_pattern, x=0, y=0,
                                    background=Color.BLACK)

    return output_grid
```

## C.3  ARC PROBLEM 2072ABA6

```python
from common import *

import numpy as np
from typing import *

# concepts:
# color mapping, pattern replication
```

```
# description:
# In the input you will see a 3x3 grid of colored pixels.
# The colors are either black or gray. The output should be a grid where:
# 1. If the pixel is black, it remains black in the output.
# 2. If the pixel is gray, it should be replaced with a 2x2 block of blue
                                    and red pixels in a checkerboard
                                    pattern.
# The blue pixel should occupy the top-left and bottom-right of the 2x2
                                    block, while the red pixel occupies
                                    the top-right and bottom-left.

def transform(input_grid):
    # Create an output grid that is larger than the input grid
    output_grid = np.zeros((input_grid.shape[0] * 2, input_grid.shape[1]
                                    * 2), dtype=int)

    # Fill the output grid based on the input grid
    for x in range(input_grid.shape[0]):
        for y in range(input_grid.shape[1]):
            if input_grid[x, y] == Color.BLACK:
                # Black stays black in the output
                continue
            else:
                # Replace gray with a checkerboard pattern of blue and
                                    red
                output_grid[2 * x:2 * x + 2, 2 * y:2 * y + 2] = [
                    [Color.BLUE, Color.RED],
                    [Color.RED, Color.BLUE]
                ]

    return output_grid
```

## C.4   ARC PROBLEM EF26CBF6

```
from common import *

import numpy as np
from typing import *

# concepts:
# color transformation, grid sections, boundary detection

# description:
# In the input, you will see a grid with a pattern of yellow and blue
                                    pixels with a black background,
# and multiple colored circles (not yellow or blue) placed randomly
                                    within the grid.
# The goal is to transform the output grid by replacing all the blue
                                    pixels with the color of the closest
                                    circle
# and keeping the yellow pixels unchanged.

def transform(input_grid: np.ndarray) -> np.ndarray:
    # Create a copy of the input grid to modify
    output_grid = np.copy(input_grid)

    # Find the coordinates of the colored circles
    circle_coordinates = np.argwhere((input_grid != Color.BLACK) & (
                                    input_grid != Color.YELLOW) & (
                                    input_grid != Color.BLUE))

    # Iterate over the grid and replace blue pixels with the nearest
                                    circle color
```

```
    for x in range(input_grid.shape[0]):
        for y in range(input_grid.shape[1]):
            if input_grid[x, y] == Color.BLUE:
                # Find the closest circle
                closest_color = None
                closest_distance = float('inf')
                for circle_x, circle_y in circle_coordinates:
                    distance = abs(circle_x - x) + abs(circle_y - y)  #
                                                        Manhattan
                                                        distance

                    if distance < closest_distance:
                        closest_distance = distance
                        closest_color = input_grid[circle_x, circle_y]
                # Replace the blue pixel with the closest circle color
                output_grid[x, y] = closest_color

    return output_grid
```

## C.5 ARC PROBLEM E7639916

```
from common import *

import numpy as np
from typing import *

# concepts:
# boundary detection, color filling

# description:
# In the input grid, you will see a black background with several purple
                                        pixels forming a boundary.
# The task is to fill the area enclosed by the purple boundary with blue
                                        pixels.

def transform(input_grid):
    # Create an output grid that starts as a copy of the input grid
    output_grid = np.copy(input_grid)

    # Find the coordinates of the purple pixels
    purple_coords = np.argwhere(input_grid == Color.PURPLE)

    # If no purple pixels are found, return the original grid
    if len(purple_coords) == 0:
        return output_grid

    # Get the boundaries of the purple pixels
    min_x, min_y = np.min(purple_coords, axis=0)
    max_x, max_y = np.max(purple_coords, axis=0)

    # Fill the area enclosed by the purple boundary
    for x in range(min_x, max_x + 1):
        for y in range(min_y, max_y + 1):
            # Check if the current position is outside the purple
                                                boundary
            if (x == min_x or x == max_x or y == min_y or y == max_y) and
                                                output_grid[x, y] ==
                                                Color.BLACK:
                output_grid[x, y] = Color.BLUE

    return output_grid
```

## C.6 ARC PROBLEM C074846D

```python
from common import *

import numpy as np
from typing import *

# concepts:
# rotation, color change, symmetry

# description:
# In the input, you will see a colored object with a single gray pixel.
# To make the output, rotate the object 90 degrees clockwise around the
#                                 gray pixel,
# and color the newly exposed pixels green.

def transform(input_grid):
    # Find the gray pixel location
    gray_pixel_locations = np.argwhere(input_grid == Color.GRAY)
    assert len(gray_pixel_locations) == 1
    gray_x, gray_y = gray_pixel_locations[0]

    # Create an output grid
    output_grid = np.full(input_grid.shape, Color.BLACK)

    # Rotate the object around the gray pixel
    for x in range(input_grid.shape[0]):
        for y in range(input_grid.shape[1]):
            if input_grid[x, y] != Color.BLACK and input_grid[x, y] !=
                                                  Color.GRAY:
                # Calculate new position after 90 degrees clockwise
                #                                 rotation
                new_x = gray_x + (y - gray_y)
                new_y = gray_y - (x - gray_x)

                # Check if the new position is within bounds
                if 0 <= new_x < output_grid.shape[0] and 0 <= new_y <
                                                  output_grid.shape[1]
                                                  :
                    # Place the rotated pixel in the output grid
                    output_grid[new_x, new_y] = input_grid[x, y]

                # Color newly exposed pixels green
                if output_grid[x, y] == Color.BLACK:
                    output_grid[x, y] = Color.GREEN

    # Place the gray pixel back in the center
    output_grid[gray_x, gray_y] = Color.GRAY

    return output_grid
```

## C.7 ARC PROBLEM AE58858E

```python
from common import *

import numpy as np
from typing import *

# concepts:
# object detection, color change, size comparison

# description:
# In the input, you will see a grid with red objects of various sizes.
# To make the output, change all objects larger than a certain size (3
#                                 pixels) to pink.
```

```python
def transform(input_grid):
    # Create a copy of the input grid to produce the output
    output_grid = np.copy(input_grid)

    # Find all connected components (red objects) in the input grid
    red_objects = find_connected_components(input_grid, background=Color.
                                        BLACK, monochromatic=True)

    # Iterate through the red objects and change their color if they are
                                        larger than 3 pixels
    for obj in red_objects:
        size = np.sum(obj == Color.RED)
        if size > 3:
            output_grid[obj == Color.RED] = Color.PINK

    return output_grid
```

## C.8   ARC PROBLEM 21F83797

```python
from common import *

import numpy as np
from typing import *

# concepts:
# vertical and horizontal lines, intersection

# description:
# In the input, you will see two red pixels.
# To make the output, draw a blue square with the red pixels as corners.
# Additionally, draw a vertical and horizontal line that intersect at
                                        each red pixel.

def transform(input_grid):
    # Copy the input grid to the output grid
    output_grid = np.copy(input_grid)

    # Find the positions of the red pixels
    red_pixels = np.argwhere(input_grid == Color.RED)

    # Ensure there are exactly two red pixels
    assert len(red_pixels) == 2
    (x1, y1), (x2, y2) = red_pixels

    # Sort the red pixels' positions to determine which is top-left and
                                        bottom-right
    top_left = (min(x1, x2), min(y1, y2))
    bottom_right = (max(x1, x2), max(y1, y2))

    # Draw a blue square from the top-left to the bottom-right corner
    output_grid[top_left[0]:bottom_right[0]+1, top_left[1]:bottom_right[1
                                        ]+1] = Color.BLUE

    # Draw horizontal and vertical lines through the red pixels
    for x, y in [(x1, y1), (x2, y2)]:
        output_grid[x, :] = Color.RED  # Horizontal line through red
                                        pixels
        output_grid[:, y] = Color.RED  # Vertical line through red pixels

    return output_grid
```

## C.9  ARC PROBLEM 33B52DE3

```python
from common import *

import numpy as np
from typing import *

# concepts:
# object detection, color mapping, grid transformation

# description:
# In the input, you will see a 20x20 grid containing a 4x4 pattern made
#                                     of colored pixels (not necessarily
#                                     contiguous) and a series of 3x3 gray
#                                      squares.
# The task is to extract the colors from the 4x4 pattern and color the
#                                     corresponding 3x3 gray squares with
#                                     those colors in a specific order.
# The first gray square should be colored with the first color from the
#                                     pattern, the second gray square with
#                                      the second color, and so on.

def transform(input_grid):
    # Step 1: Detect the 4x4 color pattern in the input grid.
    color_pattern = detect_objects(input_grid, monochromatic=False,
                                    connectivity=8)

    # Step 2: Extract the colors from the detected pattern.
    colors = []
    for obj in color_pattern:
        cropped_obj = crop(grid=obj, background=Color.BLACK)
        colors.extend(cropped_obj.flatten())

    # Step 3: Identify the locations of the gray squares.
    gray_squares = detect_objects(input_grid, colors=[Color.GRAY],
                                   monochromatic=True, connectivity
                                   =8)

    # Step 4: Color the gray squares with the corresponding colors from
    #                                 the pattern.
    output_grid = np.copy(input_grid)
    for i, gray_square in enumerate(gray_squares):
        if i < len(colors):
            # Color the gray square with the corresponding color
            x, y = np.where(gray_square == Color.GRAY)
            flood_fill(output_grid, x[0], y[0], colors[i])

    return output_grid
```

## C.10  ARC PROBLEM 332EFDB3

```python
from common import *

import numpy as np
from typing import *

# concepts:
# borders, alternating patterns

# description:
# In the input, you will see an empty black grid. To make the output,
#                                 draw a blue border around the grid,
#                                 but only on the edges that are
```

```
                                                    multiples of 2 (i.e., the top and
                                                    bottom edges, and the left and right
                                                      edges at every even column/index).

def transform(input_grid):
    # Get the dimensions of the input grid
    height, width = input_grid.shape

    # Create the output grid, initially a copy of the input grid
    output_grid = input_grid.copy()

    # Draw the blue border on edges that are multiples of 2
    for x in range(0, height, 2):  # Vertical edges: multiples of 2
        draw_line(output_grid, x, 0, direction=(0, 1), color=Color.BLUE)
                                        # Top edge
        draw_line(output_grid, x, width - 1, direction=(0, 1), color=
                                        Color.BLUE)  # Bottom edge

    for y in range(0, width, 2):  # Horizontal edges: multiples of 2
        draw_line(output_grid, 0, y, direction=(1, 0), color=Color.BLUE)
                                        # Left edge
        draw_line(output_grid, height - 1, y, direction=(1, 0), color=
                                        Color.BLUE)  # Right edge

    return output_grid
```

## D  INDUCTION FALSE POSITIVE SOLUTIONS

### D.1  ARC PROBLEM 25094A63

```
from common import *

import numpy as np
from typing import *

# concepts:
# grid transformation, color replacement, pattern detection

# description:
# In the input, you will see a grid with colored regions. If a 5x5 square
                                     in the grid consists entirely of a
                                     single color,
# replace that square with a 5x5 yellow square in the output grid. If the
                                     square is not fully filled with a
                                     single color, leave it as is.

def transform(input_grid):
    # Plan:
    # 1. Iterate over the grid to find 5x5 squares.
    # 2. For each 5x5 square, check if it is monochromatic (all pixels
                                       are the same color).
    # 3. If it is, replace it with a 5x5 yellow square in the output grid
                                       .

    output_grid = np.copy(input_grid)

    # Dimensions
    square_size = 5
    for x in range(len(input_grid) - square_size + 1):
        for y in range(len(input_grid[0]) - square_size + 1):
            # Extract the 5x5 region
            region = input_grid[x:x + square_size, y:y + square_size]
            # Check if the region is monochromatic
```

```
            if np.all(region == region[0, 0]):  # All pixels should be
                                                    the same color
                # Replace the region with a yellow square
                output_grid[x:x + square_size, y:y + square_size] = Color
                                                    .YELLOW

    return output_grid
```

## D.2  ARC PROBLEM 009D5C81

```
from common import *

import numpy as np
from typing import *

# concepts:
# pattern recognition, color transformation, grid manipulation

# description:
# In the input, you will see a 15x15 grid with a pattern of purple pixels
                                    that forms a shape and a blue
                                    circle in the center of the grid.
# To create the output, you should color the shape formed by the purple
                                    pixels with a specific color based
                                    on the number of purple pixels:
# - If there are 1-10 purple pixels, color the shape yellow.
# - If there are 11-20 purple pixels, color the shape green.
# - If there are 21-30 purple pixels, color the shape red.
# - If there are more than 30 purple pixels, color the shape orange.
# The blue circle should be replaced by black.

def transform(input_grid):
    # Create an output grid initialized to black
    output_grid = np.zeros_like(input_grid)

    # Count the number of purple pixels
    purple_count = np.count_nonzero(input_grid == Color.PURPLE)

    # Determine the color based on the number of purple pixels
    if 1 <= purple_count <= 10:
        shape_color = Color.YELLOW
    elif 11 <= purple_count <= 20:
        shape_color = Color.GREEN
    elif 21 <= purple_count <= 30:
        shape_color = Color.RED
    else:
        shape_color = Color.ORANGE

    # Find the purple shape and color it accordingly
    output_grid[input_grid == Color.PURPLE] = shape_color

    # Replace the blue circle with black
    output_grid[input_grid == Color.BLUE] = Color.BLACK

    return output_grid
```

## D.3  ARC PROBLEM E95E3D8E

```
from common import *

import numpy as np
from typing import *
```

```python
# concepts:
# occlusion, symmetry detection, reconstruction

# description:
# In the input you will see a grid containing a repeated pattern that has
                                  been partially occluded by black
                                  squares.
# To make the output, remove the occluding black squares and restore the
                                  pattern based on the detected
                                  symmetry.

def transform(input_grid):
    # Plan:
    # 1. Extract the occluded pattern from the input grid.
    # 2. Detect the symmetries in the pattern.
    # 3. Fill in the missing parts of the pattern using the detected
                                      symmetries.

    # Step 1: Identify non-black pixels to find the pattern
    occlusion_color = Color.BLACK
    non_black_mask = input_grid != occlusion_color
    objects = detect_objects(input_grid, background=occlusion_color,
                                      monochromatic=False,
                                      connectivity=8)

    # Step 2: Find the symmetries in the detected objects
    symmetries = detect_translational_symmetry(input_grid, ignore_colors=
                                      [occlusion_color])

    # Step 3: Fill in the missing parts using the symmetries
    output_grid = input_grid.copy()
    for x, y in np.argwhere(non_black_mask):
        for symmetric_x, symmetric_y in orbit(input_grid, x, y,
                                          symmetries):
            if output_grid[symmetric_x, symmetric_y] == occlusion_color:
                output_grid[symmetric_x, symmetric_y] = input_grid[x, y]

    return output_grid
```

### D.4 ARC PROBLEM 73CCF9C2

```python
from common import *
import numpy as np
from typing import *

# concepts:
# pattern extraction, rotation, centering

# description:
# In the input, you will see a grid with a pattern repeated several times
                                  , but with random rotations.
# To make the output, extract one instance of the pattern and rotate it
                                  back to its original orientation.

def transform(input_grid: np.ndarray) -> np.ndarray:
    # Create a copy of the input grid for the output
    output_grid = np.copy(input_grid)

    # Find connected components to extract patterns
    components = find_connected_components(output_grid, background=Color.
                                      BLACK, connectivity=8)
```

```python
    # For each component, check if it's a valid pattern
    for component in components:
        # Crop the component to get the pattern
        pattern = crop(component, background=Color.BLACK)

        # Check if the pattern can be rotated back to its original
                                            orientation
        # We can rotate the pattern and compare with the original
        original_pattern = pattern.copy()

        # Rotate the pattern in 90-degree increments and check if the
                                            original pattern matches
        rotated_patterns = [np.rot90(original_pattern, k) for k in range(
                                            4)]

        # We assume the first valid pattern is the original orientation
        for rotated in rotated_patterns:
            if np.array_equal(rotated, original_pattern):
                # If it matches, we've found the original orientation
                output_grid = pattern
                break

    return output_grid
```

## D.5 ARC PROBLEM D56F2372

```python
from common import *
import numpy as np
from typing import *

# concepts:
# object detection, symmetry detection, cropping

# description:
# In the input you will see a grid filled with different colored shapes.
# The output should be the smallest symmetric shape (with respect to the
                                            center) that can fit all the others
                                            inside it.
# If there are no symmetric shapes, the output should be a grid of the
                                            same size filled with the background
                                            color.

def transform(input_grid: np.ndarray) -> np.ndarray:
    # Find all connected components (shapes) in the grid
    components = find_connected_components(input_grid, background=Color.
                                            BLACK, connectivity=8,
                                            monochromatic=False)

    # Initialize a variable to track the largest symmetric shape found
    largest_symmetric_shape = None
    max_area = 0

    # Check each component for symmetry and area
    for component in components:
        # Crop the component to get its shape
        cropped_shape = crop(component, background=Color.BLACK)

        # Check for horizontal and vertical mirror symmetry
        is_symmetric = np.array_equal(cropped_shape, np.flip(
                                            cropped_shape, axis=0)) or \
                np.array_equal(cropped_shape, np.flip(
                                            cropped_shape
                                            , axis=1))
```

```python
    if is_symmetric:
        area = np.count_nonzero(cropped_shape != Color.BLACK)
        if area > max_area:
            max_area = area
            largest_symmetric_shape = cropped_shape

# If we found a symmetric shape, return it; otherwise return a grid
#                                  of the same size filled with the
#                                  background color
if largest_symmetric_shape is not None:
    return largest_symmetric_shape
else:
    return np.full(input_grid.shape, Color.BLACK)
```

# E  TEST-TIME TRAINING

Test time training is an approach for updating model parameters at test time, which we apply to our transduction model. We assume that we are given test problems $\mathcal{D}$ comprising triples $(\boldsymbol{x}_{\text{train}}, \boldsymbol{y}_{\text{train}}, x_{\text{test}})$ and a data augmentation procedure $\text{AUG}(\boldsymbol{x}, \boldsymbol{y})$ which constructs variations of an ARC problem, for example by permuting colors and rotating grids. Then model parameters are optimized to maximize the likelihood of augmented test tasks where a random training input-output is selected to serve as a fake test example:

$$\underset{\substack{(\boldsymbol{x}_{\text{train}}, \boldsymbol{y}_{\text{train}}, x_{\text{test}}) \sim \mathcal{D} \\ k \sim 1..\text{len}(\boldsymbol{x}_{\text{train}}) \\ (\boldsymbol{x}'_{\text{train}}, \boldsymbol{y}'_{\text{train}}) \sim \text{AUG}(\boldsymbol{x}_{\text{train}}, \boldsymbol{y}_{\text{train}})}}{\mathbb{E}} \left[ \mathsf{t}_\theta \left( \boldsymbol{y}_{\text{train}}[k] \mid \boldsymbol{x}_{\text{train}}[:k], \boldsymbol{x}_{\text{train}}[k+1:], \boldsymbol{y}_{\text{train}}[:k], \boldsymbol{y}_{\text{train}}[k+1:] \right) \right] \quad (6)$$

where $\text{len}(\boldsymbol{x}_{\text{train}})$ is the number of example training inputs, and $k$ is the index of the training input-output which is randomly selected to serve as a fake testcase. Note that this procedure does not rely on access to ground truth test predictions.

We choose each training example as a fake test example and do 10 times different randomly combined data augmentation for each fake task, which gives us 12k pseudo training dataset from the evaluation dataset. We also randomly include 5k RE-ARC examples and 5k ARC-Heavy examples, which we speculated would have a regularizing effect. The dataset size for test time training is 22k total problems.

# F  EXPERIMENT PARAMETERS

## F.1  INDUCTION

Fine-tuning Hyperparameters:

| training type | lora rank | lora alpha | learning rate | gradient accumulate steps |
|---|---|---|---|---|
| lora finetune | 64 | 64 | 2e-4 | 2 |

| per device batch size | device | epcoh | weight decay | learning rate scheduler type |
|---|---|---|---|---|
| 8 | 8xA100 | 3 | 0 | cosine |

For the last 230k data finetune we used full finetuning instead of LoRA:

| training type | learning rate | weight decay | gradient accumulate steps |
|---|---|---|---|
| full finetune | 1e-5 | 0.05 | 1 |

| epoch | per device train batch | learning rate scheduler type | devices |
|---|---|---|---|
| 2 | 16 | cosine | 8xA100 |

Inference Hyperparameters:

- temperature: 0.8 (1.0 for the full-data fine-tuned model)
- top-p: 1.0

Output selection: For experiments in section 4, when allowing one or two attempts, we filter the sample programs using train input-output examples and then randomly select one or two distinct programs uniformly. We report the expected value in our results. For experiments in section 5, we take the execution results of test output from the programs that can pass all the train examples and use majority vote to select the top 2, and in the case of concept arc, the top 3 test outputs.

## F.2 TRANSDUCTION

Fine-tune Hyperparameters:

| training type | learning rate | weight decay | gradient accumulate steps | device |
|---|---|---|---|---|
| full finetune | 1e-5 | 1e-2 | 2 | 8xA100 |
| engineer epoch | other epoch | per device train batch | learning rate scheduler type | |
| 3 | 2 | 8 | cosine | |

For the final engineering results, we train for 3 epochs. For all other experiment results, we train for 2 epochs.

Inference Hyperparameters:

- temperature: 0
- use beam search: True
- beam width:
    1. engineer results: 40
    2. 100k data scale: 20
    3. all other experiment results: 3
- top-p: 1.0

Test-time Fine-tuning Hyperparameters

| training type | lora rank | lora alpha | learning rate | gradient accumulate steps |
|---|---|---|---|---|
| lora finetune | 64 | 64 | 2e-4 | 2 |
| per device batch size | device | epcoh | weight decay | learning rate scheduler type |
| 2 | 4xA6000 | 3 | 0 | cosine |

