# OpenReview forum: "Combining Induction and Transduction for Abstract Reasoning"
_ICLR.cc/2025/Conference — ICLR 2025 Poster_

### Official Review · Reviewer_aaZL · 2024-10-31

**Soundness:** 3
**Presentation:** 3
**Contribution:** 3
**Rating:** 8
**Confidence:** 4

**Summary:**

The paper presents an interesting approach to combining induction and transduction for solving reasoning tasks. Induction methods predict functions that explain the training data and use them to forecast test inputs, while transduction methods directly predict test outputs based on the training and test inputs. The paper proposes a way to effectively integrate both approaches by utilizing LLMs to generate synthetic datasets, which are then used to train neural networks for induction and transduction, respectively. Experiments conducted on the ARC benchmark demonstrate that induction and transduction complement each other effectively.

**Strengths:**

Combining induction and transduction to tackle challenging reasoning tasks such as ARC appears to be a novel approach. The successful use of LLMs to generate additional examples for training the proposed model is a noteworthy demonstration. While the idea may seem straightforward, its simplicity allows for broad applicability to other tasks.

The paper is well-written and easy to follow. The technical sections are clearly explained with well-defined formulas, ensuring high-quality presentation throughout.

The discovery that induction and transduction are complementary is particularly intriguing and is likely to spur further research in the field of abstract reasoning.

**Weaknesses:**

My primary concern is that the intuition behind the proposed ensemble method isn't clearly articulated. It's not immediately evident why the approach outlined in Equations (3)-(5) is expected to be effective. When ensemble, the method starts with induction and then applies transduction if the induction doesn't find a solution—does this suggest that induction takes precedence? For example, does the method presume that any problem should be addressed with induction whenever possible? Offering more insight into the reasoning behind these choices would be beneficial. Please refer to the first question in the subsequent section as well.

While the finding regarding the complementary is interesting, it is not well-investigated why this is happening:
> line 244; most problems solved by induction are not solved by transduction, and vice versa

It would be interesting to explore why this separation occurs. What distinguishes problems suited for induction from those better addressed by transduction? Any insights into these questions would be valuable.

Minors:

As the authors noted in the limitations section, the method has only been demonstrated using the ARC benchmark. Evaluating it against other benchmarks would enhance the robustness of the paper. This raises several questions, such as whether the complementary nature of induction and transduction will apply to other reasoning benchmarks. However, I'd agree with the authors that the ARC is an unsolved, challenging task, and the experiments reasonably underpin the paper's claim.

**Questions:**

Regarding the ensemble approach:
> line135-136: Transduction and induction can be ensembled by predicting according to induction whenever at least one of the B functions fits ..., and otherwise using transduction ...

Instead of switching sequentially between these two approaches, would it be possible to develop a learning method to blend induction and transduction? A meta-function could predict whether induction or transduction should be applied based on the input. This resulting function may provide insights into what differentiates induction-friendly tasks from transduction-friendly ones.

Please refer to the weakness section for other questions.

---

> ### Author Response · Authors · 2024-11-15
>
> Thank you for your support! We really appreciate it. Below we respond to your questions and point out to where the revised draft at the top of the page has been improved in response to your input:
>
> > What distinguishes problems suited for induction from those better addressed by transduction? Any insights into these questions would be valuable.
>
> Great suggestion! We’ve added a full page of new experiments and analysis in the attached revision (Section 6). Briefly, as the revised abstract now says, “Inductive program synthesis excels at precise computations, and at composing multiple concepts, while transduction succeeds on fuzzier perceptual concepts.” Section 6 also has a new fine-grained analysis of how each model compares to human performance, finding that induction does surprisingly well on the problems that humans find hardest. Regarding human comparisons, we’ve improved our engineering so that—for the first time since ARC came out 5 years ago—we have a published model close to human level (54% vs 60%; see revised Section 5).
>
> > My primary concern is that the intuition behind the proposed ensemble method isn't clearly articulated
>
> Thanks for inviting this clarification. We’ve revised to now say the following on line 142:
> “Induction allows checking candidate hypotheses against the training examples. Therefore, we know when induction has found a plausible solution--but sometimes it fails to find any solution. Transduction has the opposite property: We can't check if its predictions match the training examples, but it always offers a candidate answer. Therefore we ensemble by attempting induction first, then transduction if none of the candidate hypotheses explained the examples”
>
> The revised introduction also provides related intuition (line 79): “induction and transduction can be trivially ensembled by using induction to generate candidate functions $f$ until either a satisfactory function is found (e.g. $f(x_\text{train})=y_\text{train}$) or until a test-time compute budget is reached, at which point, transduction kicks in as a fallback”
>
>
> > Instead of switching sequentially between these two approaches, would it be possible to develop a learning method to blend induction and transduction?
>
> Blending induction and transduction—doing both at the same time on the same problem, possibly interleaved—is precisely the next step that we hope to pursue after this paper.
>
> > A meta-function could predict whether induction or transduction should be applied based on the input. This resulting function may provide insights into what differentiates induction-friendly tasks from transduction-friendly ones.
>
> Definitely, it would give insight. We think though that precisely how induction and transduction complement each other makes an ensemble not need to predict which to use: Because induction can be validated by testing candidate hypotheses on the examples (but sometimes induction fails to make a prediction because no hypotheses work), while transduction can’t be validated (but it always makes a prediction), this means that you can simply try induction first, then transduction if it can’t find any hypotheses that fit the examples.
>
> In more general settings though, there might be noise in the examples, so you don't expect to fit them perfectly. Or you might want a prior over the hypothesis space, so you won't necessarily accept something just because it fits the data. In those cases a metamodel like you suggest would make a lot of sense.
>
> Thanks again for your support. Let us know if we can clarify anything.

---

> > ### Comment · Reviewer_aaZL · 2024-11-25
> > **Response to Authors**
> >
> > I thank the authors for their clarifications. As I do not have any major concerns after the rebuttal, I will maintain my rating (accept) and raise my confidence accordingly.

---

### Official Review · Reviewer_22bP · 2024-11-02

**Soundness:** 3
**Presentation:** 3
**Contribution:** 2
**Rating:** 6
**Confidence:** 4

**Summary:**

The paper expands ARC dataset with LLM generated synthetic data, and investigate induction and transduction approaches on this dataset.

**Strengths:**

1. The paper contributes a expanded ARC dataset generated by LLM combined with manual efforts. This is valuable to the community.
2. The paper performs a thorough evaluation of induction-based and transduction-based approach.
3. The paper is written clearly and easy to follow.

**Weaknesses:**

1. As we know LLM often hallucinates. How do the authors quality check the generated dataset?
2. While the authors concluded that induction and transduction approaches are complementary, I hoped to see more in-depth analysis of why induction works better for some while transduction works better for the others.
3. While the authors also discussed the limitation that such method is only tested on ARC, I still think this is a weakness. I wonder the conclusion would be different for some other real-world datasets.

**Questions:**

See weakness.

---

> ### Author Response · Authors · 2024-11-15
>
> Thanks for the feedback! See below for our responses, and for pointers to where we’ve revised the paper (attached at top of page) to address your concerns.
>
> > I hoped to see more in-depth analysis of why induction works better for some while transduction works better for the others
>
> Thanks for this suggestion. We’ve added a full page of new experiments and analysis in the attached revision (Section 6). Briefly, as the revised abstract now says, “Inductive program synthesis excels at precise computations, and at composing multiple concepts, while transduction succeeds on fuzzier perceptual concepts.” Section 6 also has a new fine-grained analysis of how each model compares to human performance, and we’ve improved our work so that—for the first time since ARC came out 5 years ago—we have a published model approaching human level (see revised Section 5).
>
> > LLM often hallucinates. How do the authors quality check the generated dataset?
>
> We do not use LLMs to predict program outputs, instead using a Python interpreter. Therefore, by construction, every program in the metalearning training data actually generates its corresponding input-outputs.
>
> > method is only tested on ARC
>
> ARC is a composite of hundreds of tasks. **Top conferences publish papers that evaluate solely on ARC** (ICML ‘24, [1]; NeurIPS ‘23, [2]; NeurIPS ‘22 [3]). We increase the SOTA of open methods from 15% [1] to 54% (vs. human 60%), the **first time any published method has approached human performance
> in the 5 years since ARC came out.** Frankly, we view that empirical result as being very strong—more than enough to compensate for the lack of other benchmarks.
>
> As reviewer aaZL says about only evaluating on ARC: “However, I'd agree with the authors that the ARC is an unsolved, challenging task, and the experiments reasonably underpin the paper's claim.”
>
>
> [1] ICML ‘24, CodeIt, Butt et al.
>
> [2] NeurIPS ‘23, ANPL, Huang et al.
>
> [3] NeurIPS ‘22, Communicating Natural Programs, Acquaviva et al.
>
>
> > I wonder the conclusion would be different for some other real-world datasets.
>
> For some datasets, like FlashFill-style text editing, previous research found induction superior [4]. In the attached updated paper, we found transduction superior on a different test set (ConceptARC). So yes, which method is best depends on the problems, and it speaks to ARC’s diversity that neither method clearly dominates.
>
>
> [4] ICML ‘17, RobustFill. Devlin et al.

---

> ### Author Response · Authors · 2024-11-27
>
> Dear Reviewer 22bP,
>
> Would you mind checking out our response, and letting us know if it addresses your concerns?
>
> In response to your review we've added a full page of new experiments/analysis. Also we've significantly improved the quantitative results.
>
> Thank you,
>
> Authors

---

> ### Author Response · Authors · 2024-12-02
>
> Dear reviewer,
>
> Today is the last day of reviewer discussion. Would you mind checking out our response, and new experiments?
>
> In response to your review we've added a full page of new experiments. Also we've significantly improved the quantitative results.
>
> Thank you,
> Authors

---

> ### Comment · Reviewer_22bP · 2024-12-02
> **Reply to rebuttal**
>
> The authors partially addressed my concern but I still think some more real world test sets are needed to show the impact of the proposed method. Thus I only increase the score slightly.

---

### Official Review · Reviewer_xinS · 2024-11-05

**Soundness:** 2
**Presentation:** 1
**Contribution:** 2
**Rating:** 3
**Confidence:** 3

**Summary:**

This work validated the Transduction and Induction methods on ARC tasks and found them to be suitable for different tasks. By combining these two, a methodology is proposed to gain performance improvement in ARC.

**Strengths:**

1. The authors have developed a diverse set of experiments around transduction and induction with respect to the ARC task.
2. The authors improve the performance on ARC tasks with an ensemble that includes both transduction and induction.

**Weaknesses:**

1. The overall presentation could be improved. The presentation about the main contributions is confusing. The paper's title combines induction and transduction, but the abstract claims that inductive models and transductive models perform differently. The abstract does not summarize how the authors combine induction and transduction, nor does it provide a specific statement on the differences.
2. The paper does not provide a theoretical analysis, and the experimental validation is somewhat narrow. Specifically, the authors only conducted verification on the ARC benchmark. Although the authors argue that ARC includes different types of tasks, they are all tasks in a grid-world environment, rather than real-world tasks. This casts doubt on the reliability of the proposed conclusions in the real world.
3. A more specific discussion is needed for the conclusions obtained from the authors' experimental analyses. it's reasonable that induction and transduction methods are applicable to different problems. But what exactly are the types? Figure 6 provides some examples of ARC problems, which are insufficient to give the reader valid conclusions. What we are more interested in knowing is whether there are some guidelines for determining what tasks are more suitable for induction and what tasks are more suitable for transduction.
4. The author's statement of combining induction and transduction in the title is not sufficiently elaborated in the manuscript. In Section 5, the author simply states that an ensemble method is proposed, but exactly how the two are combined is important to the current title and should be elaborated.

Overall, I think this work is more of a technical report around the ARC challenge and doesn't bring significant insights to the community.

**Questions:**

1. Are there guidelines for determining what tasks are appropriate for induction and what tasks are appropriate for transduction?
2. How is the ensemble method proposed in Section 5 designed?

---

> ### Author Response · Authors · 2024-11-15
>
> Thanks for the feedback! See below for our responses, and for pointers to where we’ve revised the paper (attached) to address your concerns.
>
> > what tasks are appropriate for induction and what tasks are appropriate for transduction?
>
> Great suggestion! We’ve added a full page of new experiments and analysis in the attached revision (Section 6). Briefly, as the revised abstract now says, “Inductive program synthesis excels at precise computations, and at composing multiple concepts, while transduction succeeds on fuzzier perceptual concepts.” Section 6 also has a new fine-grained analysis of how each model compares to human performance, finding that induction does surprisingly well on the problems that humans find hardest. Regarding human comparisons, we’ve improved our engineering so that—for the first time since ARC came out 5 years ago—we have a published model close to human level (54% vs 60%; see revised Section 5).
>
> > the author simply states that an ensemble method is proposed, but exactly how… should be elaborated
>
> We’d specified in equation 5 how the ensembling works, but we’ve revised the text to better provide intuition. The revision now says on line 142:
> “Induction allows checking candidate hypotheses against the training examples. Therefore, we know when induction has found a plausible solution--but sometimes it fails to find any solution. Transduction has the opposite property: We can't check if its predictions match the training examples, but it always offers a candidate answer. Therefore we ensemble by attempting induction first, then transduction if none of the candidate hypotheses explained the examples”
>
> The revised introduction also provides related intuition (line 79): “induction and transduction can be trivially ensembled by using induction to generate candidate functions $f$ until either a satisfactory function is found (e.g. $f(x_\text{train})=y_\text{train}$) or until a test-time compute budget is reached, at which point, transduction kicks in as a fallback”
>
>
> > The paper does not provide a theoretical analysis
>
> Thanks for raising this issue. We’ve responded with a new paragraph in the discussion on line 522:
>
> Theoretically, induction and transduction should not be so complementary. Equivalences between induction and transduction are well-know, such as the `kernel trick' which allows translating parametric function fitting into a transductive problem. Our metalearning models, given infinite metatraining data, should similarly converge because neural networks are universal function approximators. That there remains a difference is interesting precisely because it deviates from what one would expect theoretically.
>
>
> > overall presentation could be improved… The abstract does not summarize how the authors combine induction and transduction, nor does it provide a specific statement on the differences.
>
> Thanks for suggesting improving the abstract. It now includes the text:
>
> “We find inductive and transductive models solve different kinds of test problems, despite having the same training problems and sharing the same neural architecture: Inductive program synthesis excels at precise computations, and at composing multiple concepts, while transduction succeeds on fuzzier perceptual concepts. Ensembling them approaches human-level performance on ARC-AGI.”
>
>
>
>
> > the authors only conducted verification on the ARC benchmark
>
>
> ARC is a composite of many tasks. **Top conferences publish papers that evaluate solely on ARC** (ICML ‘24, [1]; NeurIPS ‘23, [2]; NeurIPS ‘22 [3]). We increase the SOTA of open methods from 15% [1] to 54% (vs. human 60%), the **first time any published method has approached human performance
> in the 5 years ARC has been out.** Frankly, we view that empirical result as being very strong—more than enough to compensate for the lack of other benchmarks.
>
>
> As reviewer aaZL says regarding only evaluating on ARC: “However, I'd agree with the authors that the ARC is an unsolved, challenging task, and the experiments reasonably underpin the paper's claim.”
>
>
> [1] ICML ‘24, CodeIt, Butt et al.
>
> [2] NeurIPS ‘23, ANPL, Huang et al.
>
> [3] NeurIPS ‘22, Communicating Natural Programs, Acquaviva et al.

---

> > ### Comment · Reviewer_xinS · 2024-11-25
> >
> > Thank you for your response.  Please see my specific response below.
> >
> >
> > **The difference between induction and transduction**
> >
> > Thank you for your effort in providing a new page of results. Nevertheless, I believe the issue remains. We can always observe some conclusions from experimental results, such as when induction works well and doesn't. However, the current analysis is provided only at a colloquial level, as you stated, "Inductive program synthesis excels at precise computations, and at composing multiple concepts, while transduction succeeds on fuzzier perceptual concepts," **without any formal analysis of the problem, which greatly diminishes the generalizability of the conclusions.** At least for now, I do not consider the conclusions I have obtained to be significant. **I suggest considering at least the following questions to make the conclusions clearer and more generalizable** to different tasks. What are "precise computations"? How are concepts combined, and what types of combinations are supported? What are "fuzzy concepts"? In existing tasks, such as ARC, which tasks demonstrate concept combination, and how are they combined? Can tasks be quantitatively compared in terms of concept combination difficulty? Can tasks be quantitatively compared in terms of concept definition fuzziness? Based on clear definitions and comparisons, the differences pointed out in this work might have the opportunity to influence how people in the field consider choosing between induction and transduction methods. Otherwise, I maintain that the current conclusion is more of a technical report.
> >
> > **Ensemble framework**
> >
> > It is reasonable that induction has higher time costs but is more likely to find precise solutions. So you use induction first, then switch to transduction after the timeout. However, what is the relationship between this and the performance differences between induction and transduction analyzed in your paper? When I started reading this paper, I had high expectations, hoping to see the authors demonstrate the differences between induction and transduction and see them make methodological choices for task reasoning based on these differences to arrive at an ensemble framework. However, **the current ensemble method shown is clearly loosely connected to the findings presented** in the paper. This further makes me feel that the current version of the work can only constitute a technical report, and there is still a significant gap before it can become a top-conference paper that influences people in the field.
> >
> > **Limited evaluation**
> >
> > I clearly perceive that the authors emphasized in their response that this is the first time in 5 years that the ARC challenge has approached human-level performance. I admit this sounds cool. Other reviewers have also mentioned that ARC is unsolved and challenging. Personally, I cannot be convinced that AGI can be achieved at the grid world level. If other reviewers all acknowledge this contribution, I can stop insisting that more experiments are necessary. Nevertheless, **please seriously consider how to generalize the conclusions from ARC to broader tasks**. You cannot expect this work to only influence people who focus on ARC. While the current discussion may tell us which tasks on ARC are more suitable for induction or transduction, for those who don't focus on this challenge, what conclusions do you want us to draw from this paper?

---

> > > ### Author Response · Authors · 2024-11-30
> > >
> > > Thank you for your engagement, and for your help in improving the paper.
> > >
> > > **Limited evaluation**
> > >
> > > > this is the first time in 5 years that the ARC challenge has approached human-level performance. I admit this sounds cool. Other reviewers have also mentioned that ARC is unsolved and challenging… **If other reviewers all acknowledge this contribution, I can stop insisting that more experiments are necessary.**
> > >
> > > Two reviewers acknowledge our contribution (the third is unresponsive). A [public commenter](https://openreview.net/forum?id=UmdotAAVDe&noteId=h7RkRV3m5h) also pushes back on your assessment.
> > >
> > > More broadly, certain benchmarks are often regarded as rich enough to single-handedly carry a paper, such as Nethack (a gridworld, yet sufficient for an [ICLR notable-top-25%-paper](https://openreview.net/pdf?id=sKc6fgce1zs)), [Omniglot](https://www.cs.cmu.edu/~rsalakhu/papers/LakeEtAl2015Science.pdf) and [Atari](https://www.nature.com/articles/nature14236) (Science and Nature), the boardgame Go (see [public comment](https://openreview.net/forum?id=UmdotAAVDe&noteId=h7RkRV3m5h)) and [yes,](https://openreview.net/pdf?id=RTRS3ZTsSj) even [ARC](https://openreview.net/pdf?id=SXVn5IFsrs), which comprises many hundreds of few-shot learning tasks. We’re not arguing other benchmarks wouldn’t be desirable: We’re simply arguing the paper shouldn’t be rejected for only benchmarking on ARC.
> > >
> > > > please seriously consider how to generalize the conclusions from ARC to broader tasks
> > >
> > > See bolded text "To what extent is this methodology applicable beyond ARC?" (lines 499-519).  We extensively discuss this *over 20 lines of text*.
> > >
> > > > I cannot be convinced that AGI can be achieved at the grid world level
> > >
> > > Strongly agree. It’s unfortunate they renamed ARC to “ARC-AGI”. We’ve revised to use the original, more humble name. The renaming was not our choice.
> > >
> > > **The difference between induction and transduction**
> > >
> > > > the current analysis is provided only at a colloquial level
> > >
> > > Not true: You quote analysis *from the abstract.* From the main paper: Section 6 gives quantitative analysis. Section 8 gives theoretical discussion under the bolded heading “Theoretically transduction and induction…”
> > >
> > > > What are "precise computations"?
> > >
> > > See line 437 (one example is ”counting”)
> > >
> > > > What are "fuzzy concepts"?
> > >
> > > See line 438 (qualitative shape relations; Fig 2 gives another example: denoising)
> > >
> > > **Ensemble framework**
> > >
> > > > you use induction first, then switch to transduction after the timeout. However, what is the relationship between this and the performance differences between induction and transduction analyzed in your paper?
> > >
> > > Theory dictates our ensemble method, which is not standard bagging/boosting/etc. Empirical practice determines it works because the models solve different problems:
> > > 1. Lines 142-147: Ensembling by running induction first, and transduction second, arises from theoretical properties of these methods *independent* of them solving different problems, but theoretically one expects better performance to the extent they solve different problems
> > > 2. Figure 5: The ensemble works in practice precisely because they indeed do solve different problems
> > >
> > > We hope this clarifies things: Theoretical properties motivate the ensemble method, but empirical performance derives from average outcomes over hundreds of learning problems.
> > >
> > > **Parting words**
> > >
> > > Obviously, we’re butting heads. We’d love to have your support, but we understand that sometimes, researchers just disagree. If you have further concerns, do let us know. If our new experiments and paper revisions have nudged you toward acceptance—even a little—we’d appreciate an acknowledgement of that as well.

---

> > > > ### Comment · Reviewer_xinS · 2024-12-02
> > > >
> > > > Thank you for your further response.
> > > >
> > > > **The difference between induction and transduction**
> > > >
> > > > The results on conceptARC (Section 6) leave the difference still unclear. How do these findings generalize to concepts in other tasks? For example, from a machine learning perspective, concept learning or combination could affect solution efficiency by influencing the hypothesis space. However, the current version doesn't propose such a hypothesis, instead focusing only on experimental numerical results. While we can always present various experimental outcomes, what matters are the conclusions we can draw from them. Simply stating that some ARC concepts are easy to learn while others are difficult is not a formalized, generalizable conclusion.  Given the current conclusions, how could researchers apply your insights when working on other tasks, such as Atari or Omniglot?
> > > >
> > > > Regarding "precise computations" and "fuzzy concepts," the concept names you presented are not formalized. Your categorization of "counting" as "precise computations" lacks generality. In fundamental machine learning terms, we optimize the model parameters $f$ in a hypothesis space $F$ using $\mathcal{X}-y$ paired datasets. What specific aspects do different concepts influence? What do "precise computations" represent in machine learning terms, and how do they contribute to Induction's advantages? A discussion of these elements may provide generalizable conclusions that the community could apply.
> > > >
> > > > **Ensemble framework**
> > > >
> > > > You argue that they solve different problems, justifying the ensemble approach. However, this doesn't clearly align with your conclusions about which types of problems each method can solve.
> > > >
> > > > ---
> > > >
> > > > Finally, thank you again for your response. It‘s unfortunate that I have not identified significant conclusions from this work, which contrasts with the high scores from two other reviewers and a public comment. I also regret that a misunderstanding of my comments prevented my concerns from being directly addressed, despite my efforts to make them as actionable as possible, following ICLR guidelines. As reviewers, our role is to provide comments and suggestions based on the paper's presentation to help authors improve the quality of their paper. In terms of the current version, I feel it has not met my expectations. I still sincerely suggest that the authors further summarize and refine their conclusions. Another reviewer has also raised questions similar to my major concerns. I believe this at least provides a perspective that could potentially complement this work.

---

> ### Public Comment · ~Yu_Sun1 · 2024-11-30
>
> Would the reviewer still have rejected this paper if the application was Go instead of ARC? All three of the reviewer's concerns could have been raise for the AlphaGo paper (Silver et al. 2016 in Nature). Specifically, AlphaGo
> - did not have theoretical analysis,
> - used many tricks that were orthogonal to their main method (MCTS with a learned value function),
> - did not evaluate on "real-world tasks" outside of the "grid world" (the Go board).
>
> These concerns are completely valid. However, rejecting the paper solely based on these concerns feels extreme.
>
> To be honest, I had the same concerns about AlphaGo when it first came out in 2016. I was especially concerned whether their findings could be relevant beyond games. However, this concern is exactly why such papers should be published, because then others can build on them. It took the community many years to build upon those findings from Go, poker, Dota and Diplomacy, before we arrived at o1. Human-level accuracy on ARC is not AGI, just like AlphaGo was not. Probably few of us would be "convinced that AGI can be achieved at the grid world level." But that doesn't mean there's nothing to learn here.

---

> ### Author Response · Authors · 2024-12-02
>
> Thanks for your continued engagement.
>
> It seems our only remaining disagreement is whether our conclusions apply to other domains. Although lines 510-520 gave various examples, but to focus just on those you mention:
>
> > how could researchers apply your insights when working on other tasks, such as Atari or Omniglot?
>
> For Omniglot we predict alphabets with counting (eg Braille) or significant compositionality (eg Hangul) favor induction. We predict cursive English favors transduction (fine-grained perception).
>
> For Atari we’d expect inductive search for policies to succeed where successful play requires precise tracking of cause-effect dependencies, like Montezuma’s revenge (counting # of keys). Transduction make sense for games like Pitfall, which demand fast perception-action mapping based on what is immediately perceived around the player. Note this *goes against conventional wisdom*, which [lumps those games together](https://tinyurl.com/Go-Explore-Nature).
>
> You’re of course correct our conclusions are empirical, and we prove no theorem separating transductive and inductive generalization (in fact, it seems theoretically unjustified: L522). But should the paper be rejected over this?
>
> Finally, we do appreciate you encouraging further discussion of how the general insights apply to other datasets. The final version will include the above discussion of Atari and Omniglot.

---

### Official Review · Reviewer_y4PU · 2024-11-06

**Soundness:** 4
**Presentation:** 4
**Contribution:** 3
**Rating:** 8
**Confidence:** 4

**Summary:**

This paper studies the differences between inductive approaches and transductive approaches on ARC tasks.  They generate a synthetic dataset of ARC problems for training.  They demonstrate that induction and transduction solve different problems, and that neither is obviously superior to the other, and rather ensembling both approaches leads to the best performance.  They achieve impressive performance (40%) using LLama 3.1.

**Strengths:**

* The study of the different types of problems that induction and transduction solve is of great conceptual interest
* The synthetic dataset and it's generation pipeline has value that can be of independent interest
* They achieve compelling performance (40%) with LLama 3.1, matching or beating stronger models GPT-4o, Claude 3.5
* The illustrations are informative

**Weaknesses:**

* A more principled understanding of the difference between induction and transduction, or a categorization of the problems they can solve  seems to be missing.  Adding some minimal insight into this could enhance the papers impact.

**Questions:**

* In the authors definition of induction, the distribution over the hypothesis $f$ depends on $x_{test}$, i.e. $f \sim i_{\theta}(x_{train}, y_{train}, x_{test})$.  In pure induction you would believe that the hypothesis only depends on the train set, i.e. $f \sim i_{\theta}(x_{train}, y_{train})$, and only at test time $x_{test}$ is used.  The authors definition seems to blur the line between induction and transduction, as their definition of induction forms the hypothesis based on the train examples and the test example, which clasically would be considered transduction, see e.g. the definition of Vapnik [1].

* How are the grids that are inputs to the ARC problem encoded.  It would be helpful to explicitly reference how in Python code the visual examples are represented, as in my understanding LLama 3.1 is a pure text model (not a multimodal model)

[1] Alex Gammerman, Volodya Vovk, and Vladimir Vapnik. Learning by transduction. preprint arXiv:1301.7375, 2013.

---

> ### Author Response · Authors · 2024-11-15
>
> Thanks a lot for your support! Below we respond to your questions and point out relevant changes in the posted revision.
>
> > a categorization of the problems they can solve seems to be missing. Adding some minimal insight into this could enhance the papers impact.
>
> Great suggestion! We’ve added a full page of new experiments and analysis in the attached revision (Section 6). Briefly, as the revised abstract now says, “Inductive program synthesis excels at precise computations, and at composing multiple concepts, while transduction succeeds on fuzzier perceptual concepts.” Section 6 also has a new fine-grained analysis of how each model compares to human performance, finding that induction does surprisingly well on the problems that humans find hardest. Regarding human comparisons, we’ve improved our engineering so that—for the first time since ARC came out 5 years ago—we have a published model close to human level (54% vs 60%; see revised Section 5).
>
>
> > The authors' definition seems to blur the line between induction and transduction, as their definition of induction forms the hypothesis based on the train examples and the test example, which classically would be considered transduction
>
> Good point! Our reasoning was that the induction model serves as a proposal distribution, so (1) in the infinite-sample limit, the exact form of the distribution shouldn’t matter, and (2) the samples are filtered (reweighed) using only the training input-outputs, not the test input. You’re of course correct though that conditioning hypothesis generation on the test input is nonstandard, and it does blur the line between induction and transduction, because we’re never *actually* in the infinite-sample limit.
>
> >  How are the grids that are inputs to the ARC problem encoded
>
> We’ve added the following text (line 157): “We encode 2D colored grids as strings using 1 token per pixel, and use newlines to delimit rows (Appendix B.1).” Appendix B.1 now reads: “We must include in our prompts for our fine-tuned models the input/output 2D colored grids of each problem. To do this we represent the problem textually by naming the colors one-by-one. We renamed certain colors which were more than one token (e.g., maroon$\to$brown saves 1 token/pixel), and presented the grid as a whitespace-delimited 2D array with newlines delimiting rows. Please see below [which shows illustrated examples]”
>
> Thanks again for your support. Let us know if we can clarify anything.

---

> > ### Comment · Reviewer_y4PU · 2024-11-21
> > **Initial response to authors**
> >
> > I thank the authors for their response.  I appreciate the addition of Section 6 which outlines different categories of problems in which induction and transduction outperform relative to each other.  I maintain my Accept rating.

---

### Author Response · Authors · 2024-11-16

Thank you everyone for the feedback and for the support. Attached is a revision containing the following key changes:

1. A full page of new analysis/experiments (Section 6) giving fine-grained insight on what problems induction and transduction are best at, and a deeper analysis of how they compare to human performance. This was suggested by all reviewers.
2. We've improved our engineering so that—for the first time since ARC came out 5 years ago—we have an open model close to human level: 54% ours vs 60% human, with the previous open SOTA at 15% and the previous closed SOTA at 42%.

Two reviewers object that we only evaluate on ARC. As reviewer aaZL says, “However, I'd agree with the authors that the ARC is an unsolved, challenging task, and the experiments reasonably underpin the paper's claim.” In our view, ARC is a composite of many tasks, which is one of the reasons why top conferences publish papers that evaluate solely on ARC (ICML ‘24, [1]; NeurIPS ‘23, [2]; NeurIPS ‘22 [3]). Given that we're releasing the first open method to approach human performance (after 5 years of researchers attempting to do so!), we think our results more than compensate for the lack of other benchmarks.

[1] ICML ‘24, CodeIt, Butt et al.

[2] NeurIPS ‘23, ANPL, Huang et al.

[3] NeurIPS ‘22, Communicating Natural Programs, Acquaviva et al.

---

### Meta-Review · Area_Chair_xX38 · 2024-12-21

**Metareview:**

This work presents a very interesting approach to combining induction and transduction on the ARC dataset; namely by detecting that the situations where one is applicable may not necessarily apply for the other. This also motivates an ensembling approach which scores well, achieving near-human-level performance on ARC.

After a detailed discussion with the Reviewers, we all agreed that this paper is presently on the borderline, with two distinct forces pulling it in opposite directions in terms of final rating:

* Focussing solely on ARC was seen as a negative aspect of the work, limiting generalisability of its conclusions.
    * I do not find this to be, in and of itself, a reason to reject. I'm well familiar with the ARC benchmark and it is certainly a major challenge on the path to AGI (being able to infer and leverage unseen reasoning mechanisms at test time, which most/all other benchmarks do not have). The results the authors achieve on ARC seem quite significant!
* Achieving a much higher score on ARC compared to baseline methods is a strongly positive aspect of the work.
    * I don't find this to be, in and of itself, a reason to accept. We decided to judge this paper for its _scientific value_, rather than the face value impressiveness of its result. An impressive result without a scientific backing or improving our collective understanding can just as easily be a blog post.

After weighing the pros and cons, the Reviewers were still not unanimous in their decision. I have decided to make a judgement call and **conditionally accept** the paper.

I do agree that ARC is a very important benchmark and any method that scores highly on it should be paid attention to -- further, the Authors take a systematic enough approach to evaluating the two directions presented on ARC that there is clear scientific value and takeaway to be had from its conclusions. I believe this makes the results ready to be shared with the ICLR community in their present form.

My **condition** for accepting the work rests in realising that a lot of this great scientific analysis is _ARC-specific_ (beyond some additional discussions on tasks like Omniglot). I would request that the Authors pre-temper any expectations about how well these claims may generalise by _modifying their title slightly_; perhaps to _"Combining Induction and Transduction for Abstract Reasoning **on ARC**"_?

I hope the Authors will be in agreement with this modification suggestion, and congratulate them on a solid contribution to reasoning research!

**Additional Comments On Reviewer Discussion:**

No additional comments beyond what is already summarised in the meta-review.

---

### Decision · Program_Chairs · 2025-01-22

Accept (Poster)